# Cyclic fasting bolsters cholesterol biosynthesis inhibitors' anticancer activity

Amr Khalifa [1,2], Ana Guijarro [1,2], Silvia Ravera[3], Nadia Bertola [3], Maria Pia Adorni[4], Bianca Papotti[5], Lizzia Raffaghello[6], Roberto Benelli[2], Pamela Becherini[1], Asmaa Namatalla[1], Daniela Verzola[1], Daniele Reverberi[2], Fiammetta Monacelli[1,2], Michele Cea [1,2], Livia Pisciotta[1,2], Franco Bernini[5], Irene Caffa [1,2,7] ✉ & Alessio Nencioni [1,2,7] ✉

Identifying oncological applications for drugs that are already approved for other medical indications is considered a possible solution for the increasing costs of cancer treatment. Under the hypothesis that nutritional stress through fasting might enhance the antitumour properties of at least some non-oncological agents, by screening drug libraries, we find that cholesterol biosynthesis inhibitors (CBIs), including simvastatin, have increased activity against cancers of different histology under fasting conditions. We show fasting's ability to increase CBIs' antitumour effects to depend on the reduction in circulating insulin, insulin-like growth factor-1 and leptin, which blunts the expression of enzymes from the cholesterol biosynthesis pathway and enhances cholesterol efflux from cancer cells. Ultimately, low cholesterol levels through combined fasting and CBIs reduce AKT and STAT3 activity, oxidative phosphorylation and energy stores in the tumour. Our results support further studies of CBIs in combination with fasting-based dietary regimens in cancer treatment and highlight the value of fasting for drug repurposing in oncology.

It is estimated that 19.3 million new cancer cases and almost 10.0 million cancer deaths occurred worldwide in 2020. In 2040, the global cancer burden is expected to be 28.4 million cases, corresponding to a 47% rise from 2020[1]. One of the greatest challenges in this scenario is represented by the cost of agents that can be used to treat cancer or prevent it in predisposed subjects, a challenge that becomes even greater in developing countries[2]. Finding uses in oncology that are outside the scope of the original medical indication of existing drugs (drug repurposing) has been proposed as a possible solution for this issue[3, 4]. Drug repurposing typically foresees reduced time and risk over de novo drug discovery and development since already-approved drugs have previously gone through the different phases of development for their original indication[3,4]. Notable examples of non-oncological drugs repurposed to treat cancer include thalidomide as a treatment for multiple myeloma and all-trans retinoic acid for acute promyelocytic leukaemia[3,5]. In addition, other already-approved agents are currently being studied in oncological clinical trials[6]. However, there remains a crucial need to identify more drugs that lend themselves to this type of use.

Studies show that short-term extreme dietary restriction or water-only fasting confer powerful antitumour activity to the antidiabetic drug, metformin, and to vitamin C (especially in tumours with mutated

[1]Department of Internal Medicine and Medical Specialties, University of Genoa, Viale Benedetto XV 6, 16132 Genoa, Italy. [2]Ospedale Policlinico San Martino IRCCS, Largo Rosanna Benzi 10, 16132 Genoa, Italy. [3]Department of Experimental Medicine, University of Genoa, Via Leon Battista Alberti 2, 16132 Genoa, Italy. [4]Department of Medicine and Surgery, University of Parma, 43125 Parma, Italy. [5]Department of Food and Drug, University of Parma, 43124 Parma, Italy. [6]Center of Translational and Experimental Myology, IRCCS Istituto Giannina Gaslini, 16147 Genoa, Italy. [7]These authors jointly supervised this work: Irene Caffa and Alessio Nencioni. ✉e-mail: irene.caffa@unige.it; alessio.nencioni@unige.it

*KRAS*[7–9]. Taken together with the results of clinical trials indicating the feasibility and overall safety of this type of dietary interventions in patients[10–13], these studies suggest that the nutritional stress imposed by fasting or fasting-based diets could be a viable approach to reveal new drug vulnerabilities and thus, help with drug repositioning in oncology.

In this work, we screen drugs that are already approved for clinical uses, searching for agents that acquire antitumour effects in the presence of culture conditions designed to recreate the metabolic effects of fasting [fasting-mimicking (FM) culture conditions, FMCC] (Fig. 1a). Using this approach, we show fasting to enhance the anti-tumour activity of several cholesterol biosynthesis inhibitors (CBIs) via reduced AKT-STAT3 signalling and oxidative phosphorylation.

## Results

### Fasting-induced drug sensitizations in cancer cells

We tested commercially available libraries of clinically approved drugs and bioactive compounds in PK9 pancreatic ductal adenocarcinoma (PDAC) cells for their effects on cell growth in the presence or absence of FMCC [i.e. 1% foetal bovine serum (FBS) and 0.5 g/L glucose][10, 14]. Using this approach, among several agents which became cytotoxic against cancer cells when combined with FMCC, we identified three fungicidal agents (imidazole derivatives), i.e. clotrimazole, miconazole, and oxiconazole (Fig. 1b and Supplementary Table 1). Azoles inhibit 14α-demethylase[15], which is a key enzyme for cholesterol biosynthesis both in *fungi* and in mammalian cells (Fig. 1c). Therefore, we evaluated whether other agents targeting this metabolic route would

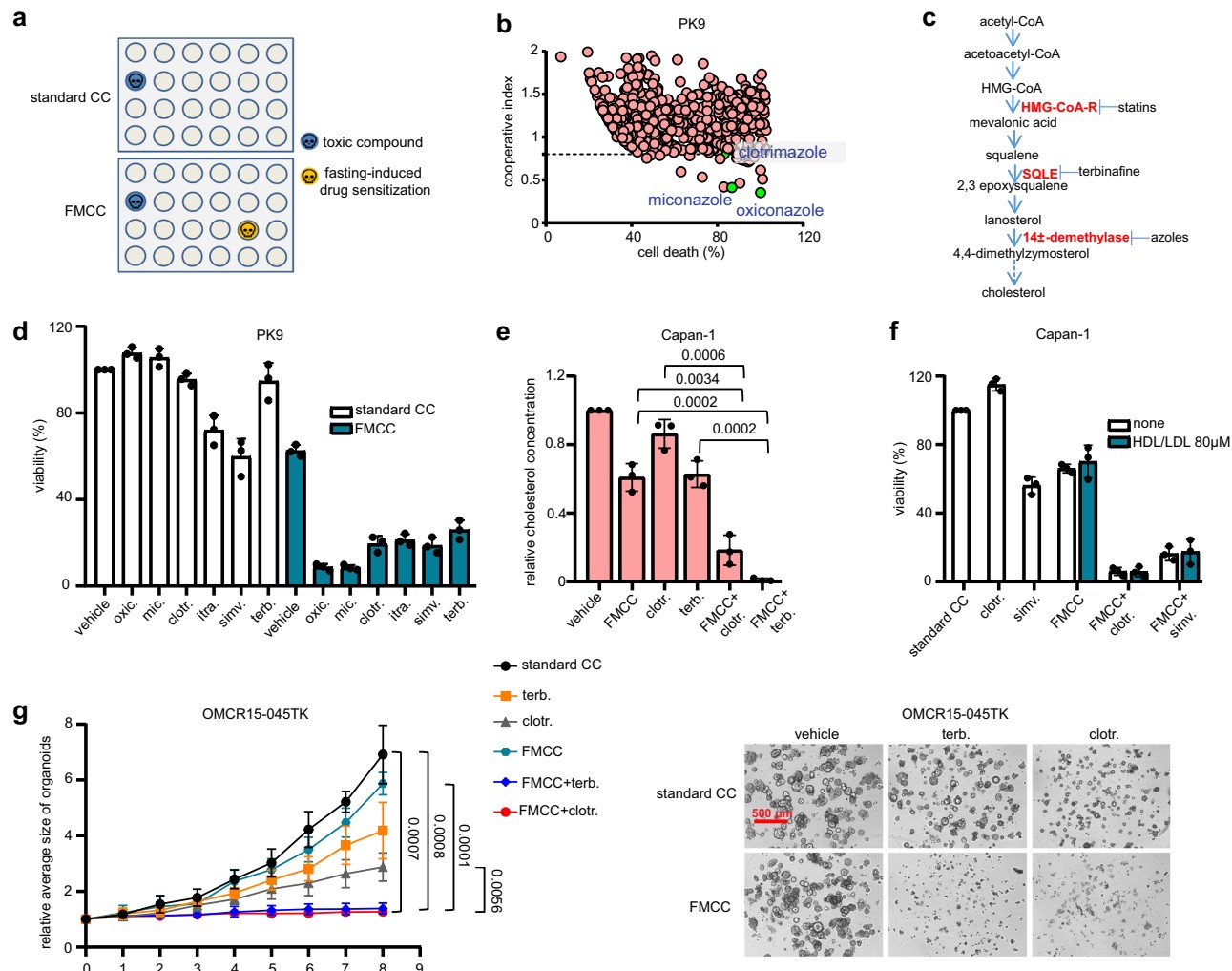

**Fig. 1 | CBIs cooperate with FMCC to kill pancreatic cancer cells and colorectal tumour organoids. a** Scheme of the screenings for agents which become cytotoxic for cancer cells through FMCC (i.e., 1% FBS and 0.5 g/l glucose). **b** PK9 PDAC cells were plated in 96-well plates and treated for 96 h w/ or w/o FMCC. During the last 72 h libraries of bioactive or clinically approved compounds were added (at a final 10 µM concentration) before cell viability was detected. A cooperative index ≤0.8 was arbitrarily chosen to select for drugs that synergistically interacted with FMCC. **c** The cholesterol biosynthesis pathway and some of its inhibitors. **d** PK9 cells were treated for 96 h w/ or w/o FMCC and for the last 72 h w/ or w/o 10 µM oxiconazole, miconazole or clotrimazole; 1 µM itraconazole; 30 µM simvastatin or terbinafine. Thereafter, cell viability was measured. **e** Capan-1 cells were treated for 24 h w/ or w/o FMCC, 25 µM clotrimazole, 50 µM terbinafine or their combination before cellular cholesterol content was measured. **f** Capan-1 cells were treated for 96 h w/ or w/o FMCC and for the last 72 h w/ or w/o 10 µM clotrimazole or 30 µM simvastatin.

Where indicated, FMCC were supplemented with 80 µM LDL/HDL cholesterol (at an LDL/HDL ratio of 3:1) to reach a final cholesterol concentration of ~90 µM in the medium. Finally, cell viability was measured. **g** Patient-derived colorectal cancer organoids (OMCR15-045TK) were treated for 192 h w/ or w/o 15 µM clotrimazole, 20 µM terbinafine, FMCC or their combination. Organoids were imaged daily. Mean organoid area was calculated from ten biological replicates using Image-J. Organoid photographs were obtained on the last day of the experiments. A scale bar is included in the upper left photograph. The drug screen (**b**) was performed once. Hit compounds (26) were retested twice; 10 of them were confirmed to synergyze with FMCC. In d-f, data points are experimental replicates; data are shown as mean ± SD. In (**g**) (organoid growth curves), data are presented as mean ± SD of three experimental replicates. *p* values were calculated by two-tailed Student's *t* test. Source data are provided as a Source Data file.

also acquire antitumour effects through fasting. Consistent with this hypothesis, we found FMCC to also confer strong cytotoxic activity to itraconazole, another 14α-demethylase inhibitor, simvastatin, a 3-hydroxy-3-methylglutaryl-CoA reductase (HMGCR) inhibitor and to the squalene epoxidase (SQLE)[16,17] inhibitors, terbinafine, butenafine and liranaftate in PK9 and Capan-1 PDAC cells and in HCT116 colorectal cancer (CRC) cells (Fig. 1d, Supplementary Fig. 1a-e). FMCC cooperated with clotrimazole and terbinafine to reduce intracellular cholesterol in cultured Capan-1 cells (Fig. 1e). Methyl-β-cyclodextrin (MβCD), which acutely depletes intracellular cholesterol[18], also had its antitumour effects enhanced by FMCC in Capan-1 PDAC cells and, when added to combined FMCC and azoles or FMCC and terbinafine, it further reduced Capan-1 cell viability (Supplementary Fig. 1f). Finally, cancer cell supplementation with a water-soluble cholesterol formulation (cholesterol-MβCD) attenuated the cooperation between FMCC and CBIs in Capan-1 and HCT116 cells (Supplementary Fig. 1g, h). Taken together, these findings indicate that FMCC and CBI's cooperate to lower intracellular cholesterol and that the antitumour effects of combined CBIs and FMCC reflect the lower cholesterol levels in cancer cells. Exposing cells to FMCC implies lowering FBS in the tissue culture medium by 90% (from 10% to 1%). As a result, cholesterol concentration in the medium will also be reduced to a similar extent. Therefore, we assessed whether FMCC-mediated enhancement of CBIs' activity reflected the reduction in cholesterol availability to cancer cells, a condition that does not occur in vivo during short-term fasting (see below) and that could artificially force cancer cells to rely on endogenous cholesterol biosynthesis. Cholesterol concentration in regular FBS is 0.9 mM[19]. Therefore, in a standard tissue culture medium, which contains 10% FBS, cholesterol concentration can be estimated to be ~90 μM. We found that FMCC retained their ability to enhance CBIs's activity even when the FMCC medium was supplemented with LDL plus HDL cholesterol (at a 3:1 ratio) to recreate a final cholesterol concentration that was similar to that found in standard cell culture medium ( ~ 90 μM; Fig. 1f, Supplementary Fig. 1i). Even when we supplemented FMCC medium with LDL and HDL cholesterol (always at a 3:1 ratio) to reach a final cholesterol concentration that was similar to that from human interstitial fluids, which represent the microenvironment bodily tissues, including cancer cells, are exposed to (i.e., ~1 mM)[20], we still found the cooperation between FMCC and CBIs to be readily detectable (Supplementary Fig. 1j). Thus, these findings indicated that FMCC-mediated enhancement of CBIs antitumour activity did not reflect reduced cholesterol availability to cancer cells through reduced FBS.

CBIs' potentiation through FMCC was not restricted to gastrointestinal cancer cells. A drug screen in A549 lung cancer cells also showed the imidazole derivatives to became cytotoxic for lung cancer cells when used in the presence of FMCC (Supplementary Fig. 2a). In addition, we found FMCC to increase the anticancer activity of CBIs in other CRC and PDAC cell lines, as well as in lung, ovarian, stomach, breast, prostate and melanoma cancer cell lines (Supplementary Fig. 2b-g and Supplementary Fig. 3a, b). Therefore, taken together, these data suggest a broad antitumour effect of this combined treatment. Finally, we tested CBIs and FMCC in CRC tumour organoids (OMCR15-045TK and OMCR16-005TK)[21] and also in these models, a striking synergy between the two interventions in terms of organoid growth inhibition was observed (Fig. 1g, Supplementary Fig. 3c and Supplementary Movie 1).

## Fasting enhance CBIs' antitumour activity in vivo

We subsequently evaluated whether periodic fasting cycles would also enhance CBIs' anticancer activity in vivo utilizing different mouse models of solid tumours. Weekly cycles of 48 h water-only fasting boosted terbinafine and clotrimazole antitumour effects in Capan-1 xenograft-bearing mice (Fig. 2a, b and Supplementary Fig. 4a, b). Consistent with the hypothesis that CBIs cooperate with fasting by

reducing cholesterol in cancer cells, we found terbinafine plus fasting to blunt the cholesterol content of Capan-1 xenografts (Fig. 2c). Combined fasting and terbinafine did not affect total serum cholesterol levels or serum HDL cholesterol in mice (Fig. 2d, e). However, this combination reduced LDL cholesterol, which is the main form of cholesterol transport from the liver to peripheral tissues, including tumours (Fig. 2e) and caused a consistent increase in the HDL/LDL ratio. Weekly fasting cycles also sensitized HCT116 xenografts to both simvastatin and clotrimazole (Fig. 2f-i). In this model, we also found that adding back LDL cholesterol in mice treated with fasting cycles plus clotrimazole abolished the enhancement of the CBI's activity through fasting and, consistently, restored intratumour cholesterol content (Fig. 2h-j). Thus, these findings are also in line with the effect of combined clotrimazole and fasting reflecting cholesterol depletion in cancer cells. We documented a marked enhancement of clotrimazole antitumour activity in immunocompetent mice, i.e. C57BL/6, in which we had established subcutaneous allografts of the melanoma cell line, B16 (Supplementary Fig. 4c). Finally, since previous studies showed the ability of fasting to synergize with several, commonly used chemotherapeutics[22], we compared the activity of the combination, fasting plus terbinafine, with that of combined fasting and oxaliplatin in Capan-1 xenograft-bearing animals. Here, we found both regimens to be very active, with essentially no difference in terms of their ability to control tumour growth over time (Supplementary Fig. 4d, e).

Altogether, these experiments confirmed that fasting and CBIs cooperate to reduce intratumour cholesterol levels and that this effect is a key determinant of the antitumour effect that is obtained by combining the two interventions.

## Toxicity of fasting, CBIs and their combination

Supplementary Table 2 shows the blood counts and the biochemistry tests that we performed in animals receiving clotrimazole, weekly 48 h fasting or their combination w/ or w/o add back of LDL-cholesterol (Fig. 2h), while Supplementary Table 3 shows the blood tests we performed in mice treated with simvastatin, weekly fasting or their combination (Fig. 2f). In both experiments, fasting caused an increase in serum aspartate aminotransferase (AST), a marker of liver damage. Combined fasting and simvastatin raised both the levels of AST and of alanine aminotransferase (ALT), another marker of acute liver damage, while in response to fasting plus clotrimazole, a significant increase in AST could not be documented. In both experiments, fasting, but not fasting plus the CBI, was also associated with higher serum levels of creatine kinase (CK), a marker of muscle injury/myopathy. Fasting, but not fasting plus clotrimazole, reduced total white blood cells, neutrophils and lymphocytes (Supplementary Table 2). The clinical parameters that we routinely monitored to detect pain or distress in mice (i.e. behaviour, posture, bodily functions, weight loss) failed to show signs of higher toxicity of fasting plus a CBI as compared to just weekly fasting. For instance, no worsening of fasting-induced weight loss could be detected with the concomitant administration of fasting plus a CBI as compared with just fasting and neither did adding a CBI to fasting affect mouse ability to re-gain weight after the fasting cycles (Supplementary Fig. 5a, b).

We also evaluated potential toxicities of fasting, CBIs and their combination in vitro, using non-malignant human cells, such as peripheral blood mononuclear cells (PBMCs) which mostly consist of CD3+ T lymphocytes, and the pancreatic ductal epithelial cell line, HPNE. As compared to resting/unstimulated PBMCs, PBMCs that were stimulated with the mitogen phytohemoagglutinin (PHA) showed higher sensitivity to FMCC, becaming more apoptotic in response to these conditions (Supplementary Fig. 5c). However, neither in resting nor in activated PBMCs, did we detect a cooperation between FMCC and CBIs, including simvastatin, in terms of ability to induce apoptosis. HPNE cells also proved very sensitive to FMCC, since these culture conditions reduced cell viability by ~60% (Supplementary Fig. 5d).

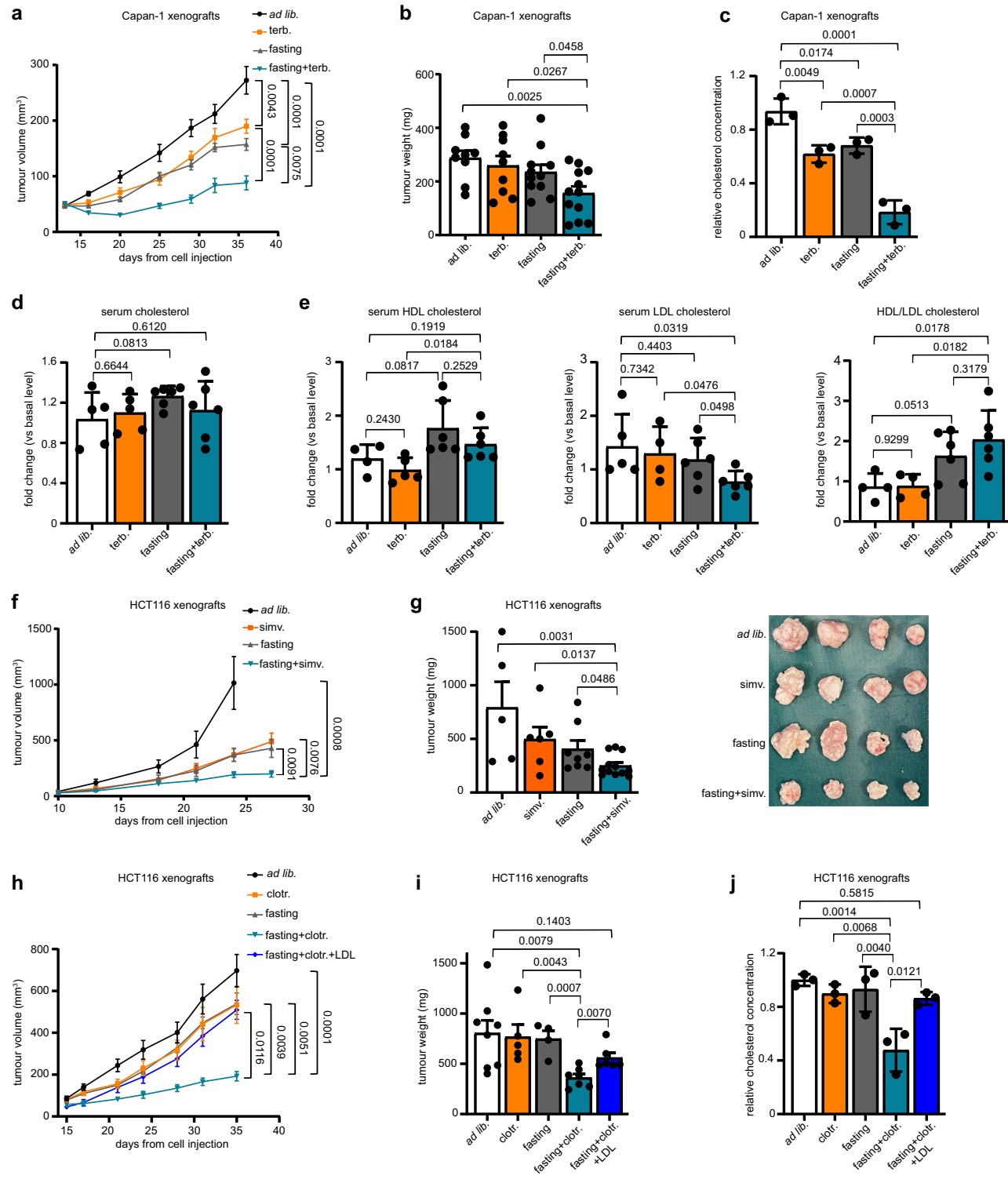

However, neither the chemotherapeutic, oxaliplatin, nor the CBIs acted synergistically with FMCC to reduce cell viability. Instead, the interaction between drugs and FMCC was mostly either additive or even infra-additive in these cells (Supplementary Fig. 5d).

**Regulation of tumour cholesterol metabolism by fasting**

We subsequently focused on the mechanisms through which FMCC (in vitro) and fasting (in vivo) enhanced the anticancer effects of CBIs. Specifically, we first assessed whether FMCC would make CBIs more active in cancer cells through the reduction in glucose or, rather, in

growth factors. Restoring normal glucose levels in cell culture did not have any effect on the potentiation of clotrimazole or terbinafine activity via FMCC (Fig. 3a). Conversely, restoring serum levels to those present in standard culture conditions (10% FBS) completely abolished the cooperation between FMCC and CBIs (Fig. 3a). We verified that in mice, fasting with or without clotrimazole reduced Igf1, c-peptide (a marker of insulin production) and leptin (Fig. 3b), three fasting-reduced factors (FRFs) which we and others previously found to partake in fasting-mediated enhancement of the activity of chemotherapy and of endocrine therapies for breast cancer[10,22,23]. Similar to serum

**Fig. 2 | In vivo activity of cyclic fasting and CBIs against PDAC and CRC xenografts. a**–**e** Capan-1 xenografts were established in 6-8-week-old female athymic nude mice. Once tumours were palpable, mice were randomized to be treated with *ad libitum* diet (*n* = 12), weekly 48 h water-only fasting ("fasting"; *n* = 12), terbinafine (40 mg/kg/day for five days a week; *n* = 10) or combined fasting plus terbinafine (*n* = 12). Tumour volume was monitored at the indicated time points (**a**). At the end of the experiment, the excised tumours were weighted (**b**) and used for cholesterol content determination (**c**). Peripheral blood was collected before treatment onset and at the end of the experiment and serum was used to detect total cholesterol (**d**), HDL and LDL cholesterol (**e**) by standard biochemistry. HDL and LDL cholesterol concentration was used to determine the HDL/LDL ratio (**e**). HCT116 xenografts were established in 6-8-week-old female athymic nude mice. Once tumours were palpable, mice were randomized to be treated with *ad libitum* diet (*n* = 7 in (**f**); *n* = 8 in **h**), weekly 48 h water-only fasting (*n* = 9 in **f**; *n* = 6 in **h**), simvastatin (in **f**, 80 mg/ kg/day; *n* = 7), combined fasting plus simvastatin (in **f**, *n* = 10) clotrimazole (in **h**, 60 mg/kg twice a week; *n* = 7), combined fasting plus clotrimazole (in **h**, *n* = 8), or fasting, clotrimazole and human LDL (in **h**, 0.25 mg/mouse twice a week coupled with fasting; *n* = 6). Tumour volume was monitored at the indicated time points (**f**, **h**). At the end of the experiment, tumours were weighted (**g**, **i**), imaged (**g**) and utilized for intratumour cholesterol determination (**j**). In (**a**, **f**, **h**), *n* indicates the number of tumours per treatment group. In (**b**, **c**–**e**, **g**, **i**, **j**), data points are biological replicates: they represent single tumours (**b**, **c**, **g**, **i**, **j**) or sera from different animals (**d**, **e**). Data are shown as mean ± SEM in (**a**, **b**, **f**, **g**, **h**, **i**) and as mean ± SD in (**c**–**e**, **j**). In (**a**, **c**, **h**, **j**), data were analysed by one-way ANOVA with Tukey post-test. In (**b**, **d**, **e**, **f**, **g**, **i**), *p* values were calculated by two-tailed Student's *t* test. In (**f**), the statistical analysis was performed utilizing the tumour volumes from day 24. Source data are provided as a Source Data file.

add-back, supplementation of the culture medium with combined IGF1, insulin and leptin, but not with the single factors, blocked the enhancement of terbinafine cytotoxic activity induced by FMCC (Fig. 3c). Consistent with these results, this cocktail of proteins also abolished FMCC-mediated potentiation of terbinafine activity in terms of intracellular cholesterol depletion in cultured Capan-1 cells (Fig. 3d). Similar to LDL cholesterol add-back, administering recombinant insulin, IGF1 and leptin to Capan-1 xenograft-bearing mice abolished fasting-induced enhancement of terbinafine activity (Fig. 3e, f) and it also restored cholesterol content in the tumours (Fig. 3g). Therefore, altogether these data indicated a non-redundant role for the reduction in IGF1, insulin and leptin in the synergy between fasting and CBIs.

Concerning the mechanisms whereby the fasting-induced reduction in insulin, IGF1 and leptin mediates cancer cell sensitization to CBIs, we first focused on the expression of enzymes from the cholesterol biosynthesis pathway. This was done in view of studies showing that these circulating factors and the signalling cascades they trigger (such as the PI3K-AKT-mTOR pathway) can activate the cholesterol production pathway through various mechanisms, including increased expression and activity of sterol regulatory element-binding protein 2 (SREBP2), a transcription factor that promotes the expression of genes encoding for cholesterol-producing enzymes[24, 25]. In Capan-1 xenografts, we found the expression of several enzymes involved in cholesterol biosynthesis, such as 3-hydroxy-3-methylglutaryl-CoA synthase 1 (*HMGCS1*), isopentenyl-diphosphate delta isomerase 1 (*IDI-1*), farnesyl diphosphate synthase (*FDPS*), lanosterol synthase (*LSS*), 24-dehydrocholesterol reductase (*DHCR24*) and 7-dehydrocholesterol reductase (*DHCR7*) to be reduced in response to fasting and/or to fasting plus terbinafine (Fig. 4a). The add-back of insulin, IGF1 and leptin prevented *HMGCS1*, *IDI-1*, *FDPS* and *LSS* downregulation occurring in response to fasting plus terbinafine, whereas in the case of *DHCR24* and *DHCR7* expression, we could not detect a significant rescue effect in response to the addback of the three factors. We also readily detected downregulated DHCR24 and SQLE expression in protein lysates from HCT116 xenografts which were treated with weekly fasting plus clotrimazole (Fig. 4b). We could not demonstrate any effect of fasting or fasting plus terbinafine on LDL receptor expression, both at the mRNA and at the protein level (Fig. 4c, d), suggesting that LDL receptor-mediated cholesterol uptake is unlikely to be affected by these interventions. Altogether, these findings indicate that several cholesterol-producing enzymes become downregulated in tumour xenografts after treatment with fasting and/or with fasting combined with a CBI and that the fasting-induced reduction in circulating insulin, IGF1 and leptin takes part in the downregulation of at least some of these enzymes.

Not only does the intracellular cholesterol content depend on cholesterol biosynthesis and on the uptake of circulating cholesterol, but it also depends on the degree of cholesterol efflux from the cell[26]. This is considered to be the first step of cholesterol return from peripheral tissues to the liver for biliary excretion, a process also known as reverse cholesterol transport[26]. Non-esterified cholesterol is effluxed from cells to extracellular HDLs through the action of transporters, such as ATP Binding Cassette Subfamily A Member 1 (ABCA1) and ATP Binding Cassette Subfamily G Member 1 (ABCG1)[26]. Specifically, ABCA1 mediates the secretion of cellular free cholesterol to the extracellular acceptor, apolipoprotein AI (APO-AI), to form nascent HDL, while ABCG1 promotes free cholesterol efflux to mature HDL. Studies show that insulin[27–29], IGF1[30–32] and leptin[33] all have the ability to negatively affect cholesterol efflux. In this context, insulin was shown to dampen ABCG1 expression and to down-regulate ABCA1 activity through its specific phosphorylation[28, 29]. Leptin was reported to accelerate cholesteryl ester accumulation in human macrophages by increasing ACAT1 expression via JAK2 and PI3K, thereby suppressing HDL-mediated cholesterol efflux[33]. In Capan-1 xenografts, we found weekly fasting and fasting plus terbinafine to increase ABCG1 expression both at the mRNA and at the protein level (Fig. 5a, b). However, when insulin, IGF1 and leptin where administered to the mice receiving fasting plus terbinafine, *ABCG1* upregulation was abolished (Fig. 5a). No increase in ABCA1 expression in response to these treatments was observed (Fig. 5b). A higher *ABCG1* expression in response to fasting plus simvastatin was also detected in HCT116 xenografts treated with simvastatin plus fasting (Fig. 5c). In the latter model, combined fasting and simvastatin also downregulated *ACAT1* expression (Fig. 5c). Consistent with the in vivo evidence that fasting stimulates ABCG1, but not ABCA1 expression, we observed that in vitro, cholesterol efflux from both HCT116 and MCF7 cancer cells only increased under FMCC when HDLs, but not ApoA-I were used as acceptors (Fig. 5d). Moreover, when the FMCC medium was supplemented with insulin, IGF1 and leptin, this enhancement in cholesterol efflux to HDL through FMCC was less pronounced as compared to regular FMCC (Fig. 5d). Therefore, these findings are consistent with fasting promoting cholesterol efflux in these cancer cells through reduced insulin, IGF1 and leptin levels. Finally, in line with the ability of fasting to increase ABCG1 expression in tumours in vivo and thus, conceivably, to enhance cholesterol efflux to mature HDLs, we found the administration of HDLs to Capan-1 xenograft-bearing mice to increase fasting's ability to reduce intratumour cholesterol and to slow tumour growth (Fig. 5e–g). Overall, these results indicate that fasting promotes cholesterol efflux via ABCG1 and possibly also via reduced ACAT1 expression, at least in some types of tumour cells. In turn, fasting-induced cholesterol efflux is likely to contribute to lower cholesterol content inside cancer cells.

## Combined fasting and CBIs blunt AKT and STAT3 activity

In subsequent experiments, we studied the effects of cholesterol depletion via fasting, CBIs and their combination on the main signalling cascades controlling cancer cell proliferation and survival, such as PI3K-AKT, Janus kinase (JAK)-STAT, and MAP kinase signalling[34]. Combined treatment with CBIs and fasting blunted AKT phosphorylation in Capan-1 and HCT116 tumour xenografts, as well as in CRC tumour organoids (Fig. 6a and Supplementary Fig. 6a–c). In HCT116

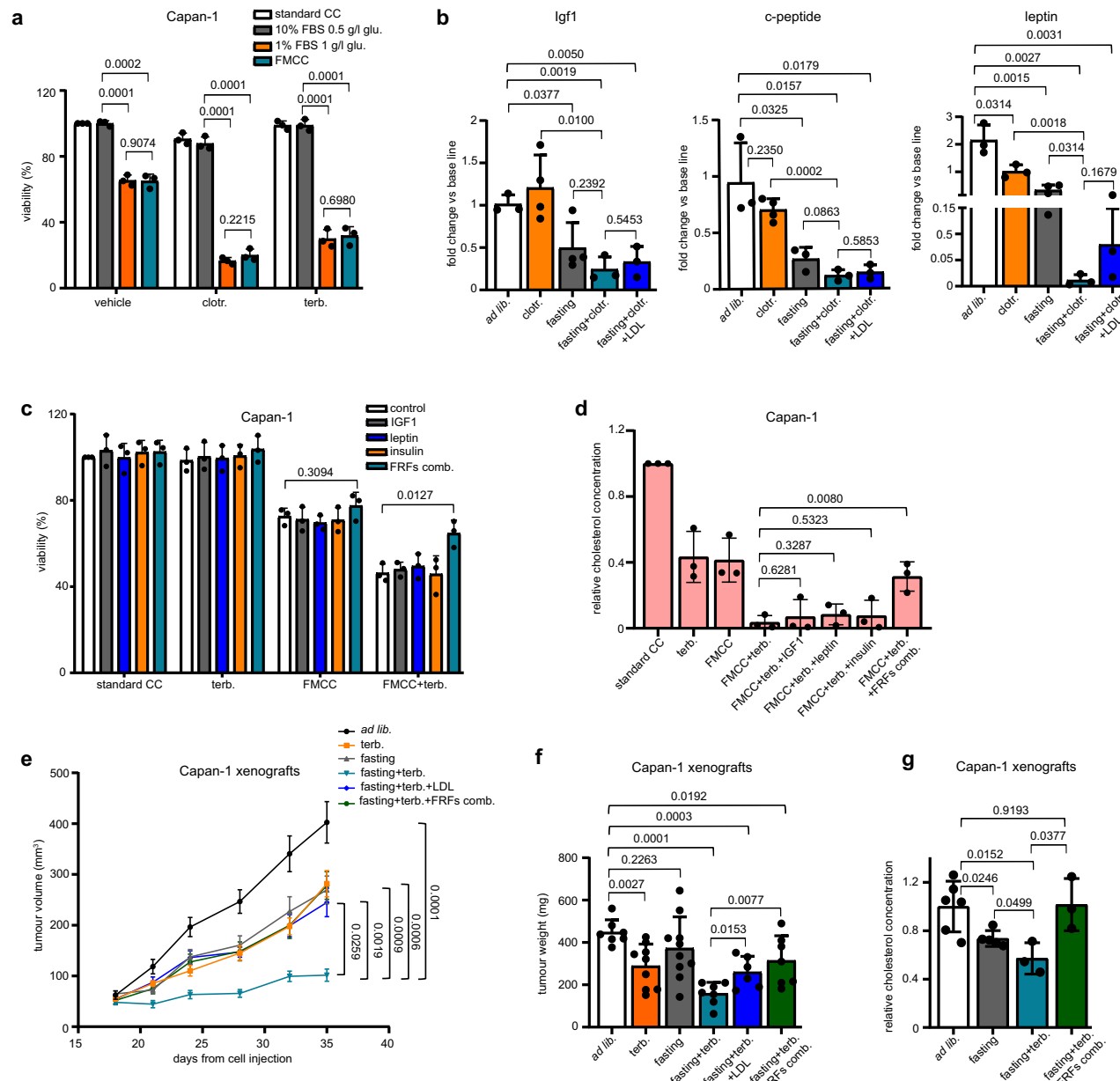

**Fig. 3 | Fasting and CBIs cooperate to slow tumour growth by reducing circulating insulin, IGF1 and leptin. a** Capan-1 cells were treated for 96 h with standard FBS (10%) and glucose (1 g/L) concentration (standard CC), standard FBS, but low glucose (0.5 g/L) concentration, low FBS (1%), but standard glucose concentration, or FMCC (1% FBS, 0.5 g/L glucose). During the last 72 h, cells were also stimulated w/ or w/o 15 µM clotrimazole or 30 µM terbinafine. Thereafter, cell viability was determined. **b** Serum was collected before treatment onset and at the end of the experiment from the animals presented in Fig. 2h (HCT116 xenografts) and Igf1, c-peptide and leptin concentrations were measured by ELISA. **c** Capan-1 cells were treated for 96 h w/ or w/o FMCC and for the last 72 h w/ or w/o 20 µM terbinafine, IGF1 (5 ng/ml), leptin (50 ng/ml), insulin (500 pM), or IGF1+leptin+insulin ("fasting-reduced factors" combination, FRFs comb.). Thereafter, cell viability was measured. **d** Capan-1 cells were treated for 48 h w/ or w/o FMCC, IGF1 (5 ng/ml), leptin (50 ng/ml), insulin (500 pM), or IGF1+leptin+insulin. 20 µM terbinafine was added during the last 24 h. Thereafter, cellular cholesterol content was measured. **e**–**g** Capan-1

xenografts were established in 6-8-week-old female athymic nude mice. Once tumours were palpable, mice were randomized to be treated with *ad lib.* diet ($n = 8$), weekly 48 h water-only fasting ($n = 11$), terbinafine ($n = 11$), combined fasting plus terbinafine ($n = 12$), fasting plus terbinafine and human LDL ($n = 8$) or fasting, terbinafine and combined intraperitoneal (i.p.) injection of insulin, IGF1 and leptin ("fasting-reduced factors" combination, FRFs comb.; $n = 13$). Tumour volume was determined at the indicated time points (**e**). At the end of the experiment, tumours were weighted (**f**) and used for cholesterol content determination (**g**). In (**e**), $n$ indicates the number of tumours per treatment group. In (**a**, **c**, **d**), data points are experimental replicates. In (**b**, **f**, **g**), data points are biological replicates: they represent sera from different animals (**b**) or single tumours (**f**, **g**). In (**a**–**d**, **f**–**g**) data are shown as mean ± SD and $p$ values were calculated by two-tailed Student's *t* test. In (**e**), data are shown as mean ± SEM and $p$ values were determined by one-way ANOVA with Tukey post-test. Source data are provided as a Source Data file.

xenografts, AKT inhibition via combined fasting plus CBIs was prevented by LDL cholesterol add-back, thus confirming that reduced AKT activation in response to the combined treatment was caused by reduced intratumour cholesterol (Supplementary Fig. 6a). Cholesterol plays a key role in lipid raft integrity, which in turn, is required for pro-

oncogenic signalling at this level, including PI3K-AKT signalling[35]. Consistent with this notion and with the cooperation between CBIs and fasting in reducing intracellular cholesterol, after isolating lipid rafts from HCT116 cells treated with clotrimazole, FMCC or their combination, we found a marked decrease in phosphorylated AKT at

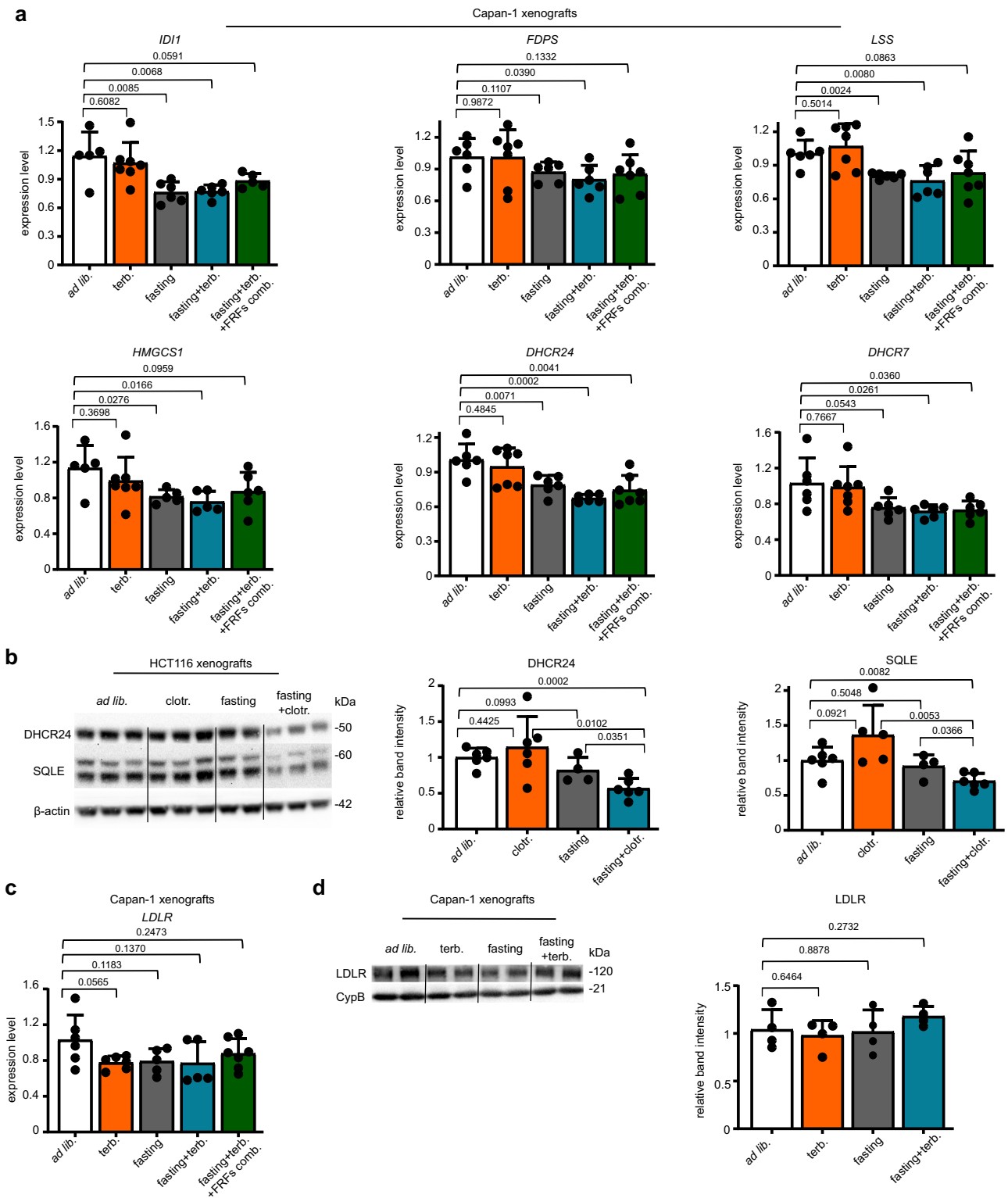

the level of these membrane microdomains in response to combined FMCC and CBI (Supplementary Fig. 6d). Supplementation of the culture medium with insulin, IGF1 and leptin avoided AKT inhibition by terbinafine plus FMCC in Capan-1 cells (Supplementary Fig. 6e), which is in line with the ability of these proteins to prevent the enhancement of terbinafine's activity in cancer cells by fasting in terms of cell viability, cholesterol content and in vivo growth (Fig. 3c–g). Restoration of AKT phosphorylation by insulin, IGF1 and leptin in cells treated with terbinafine plus FMCC was prevented by intracellular cholesterol

depletion with MβCD (Fig. 6b), supporting the notion that the effect of supplemented insulin, IGF1 and leptin on AKT activation (in cells treated with FMCC plus CBIs) reflects their ability to increase cholesterol content inside the cancer cell. Capan-1 and PANC-1 PDAC cells transduction with constitutively active, myristoylated AKT (myr-AKT)[10] conferred significant protection from combined CBIs and FMCC (Supplementary Fig. 6f). Similar results were obtained in MCF7 breast cancer cells (Supplementary Fig. 6g). Altogether, these experiments demonstrated that fasting cooperates with CBIs to reduce AKT

**Fig. 4 | Fasting and CBIs cooperate to reduce the expression of enzymes from the cholesterol biosynthesis pathway. a** Tumours (Capan-1 xenografts) were isolated at the end of the experiment presented in Fig. 3e and utilized for RNA isolation and to quantify the expression of the cholesterol-producing enzymes *IDI1, FDPS, LSS, HMGCS1, DHCR24* and *DHCR7* by QPCR. "FRFs comb." indicates the combined administration of the factors, insulin, IGF1 and leptin ("fasting-reduced factors combination"). **b**, Tumours (HCT116 xenografts) were isolated at the end of the experiment presented in Fig. 2h and utilized for protein lysate generation and for the subsequent detection of DHCR24, SQLE and β-actin by Western blotting. The intensity of the DHCR24 and SQLE bands was quantified and normalized to that of β-actin bands. **c** *LDLR* expression was determined by QPCR in tumours (Capan-1

xenografts) that were isolated at the end of the experiment presented in Fig. 3e. **d** Protein lysates were generated from tumours (Capan-1 xenografts) isolated at the end of the experiment presented in Fig. 3e and utilized for the detection of LDLR and Cyclophilin B (CypB) by Western blotting. The intensity of the LDLR bands was quantified and normalized to that of the CypB bands. The Western blots shown in (**b, d**) are one independent experiment out of two. In the Western blot band quantifications (**b, d**), samples derive from the same experiment; blots were processed in parallel. All data points from (**a–d**) are biological replicates (they represent single tumours). Data are shown as mean ± SD. *p* values were calculated by two-tailed Student's *t* test. Source data are provided as a Source Data file.

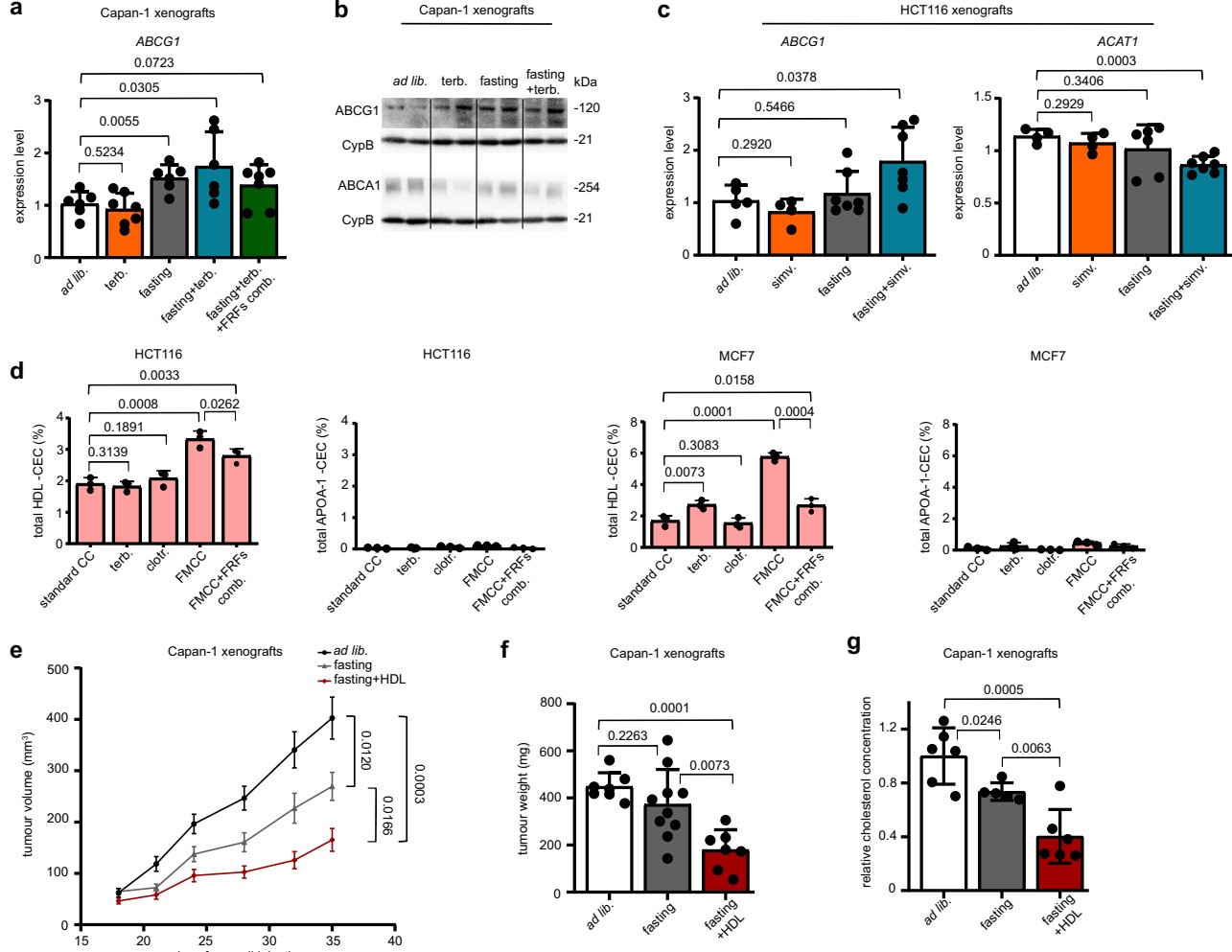

**Fig. 5 | Fasting promotes cholesterol efflux in cancer cells.** Tumours (Capan-1 xenografts) were isolated at the end of the experiment presented in Fig. 3e and utilized for RNA isolation and for the subsequent measurement of *ABCG1* expression by QPCR (**a**) or for protein lysate generation and subsequent detection of ABCG1, ABCA1 and cyclophilin B (CypB) by Western blotting (**b**). **c** Tumours were isolated at the end of the experiment presented in Fig. 2f (HCT116 xenografts) and utilized for RNA isolation and for the detection of *ABCG1* and *ACAT1* expression by QPCR. **d** HCT116 and MCF7 cells were plated in 96-well plates. Radiolabelling was carried out in standard CC w/ or w/o 20 µM terbinafine, 10 µM clotrimazole, or in FMCC w/ or w/o IGF1 (5 ng/ml)+leptin (50 ng/ml)+insulin (500 pM) ("fasting-reduced factors" combination, FRFs comb.). The cholesterol acceptors apolipoprotein A-I (10 µg/ml) or HDL (12.5 µg/ml) were added for 4 h before the medium was collected and radioactivity was quantified (CEC: cholesterol efflux capacity). **e–g** Capan-1 xenografts were established in 6-8-week-old female athymic nude mice. Once tumours were palpable, mice were randomized to be treated with *ad lib.*

diet (*n* = 8), weekly 48 h water-only fasting (*n* = 11), or fasting plus human HDL (1 mg/mouse twice a week coupled with fasting; *n* = 7). Tumour volume was calculated at the indicated time points (**e**). At the end of the experiment, tumours were weighted (**f**) and tumour cholesterol content was quantified (**g**). In (**e–g**), the data from mice that were treated with just *ad lib.* diet or with just fasting were already presented in Fig. 3e–g. They are shown here again to allow comparison with the group, fasting+HDL (all of these groups were included the same experiment). In (**e**), *n* indicates the number of tumours per treatment group. In (**a, c, f, g**) data points are biological replicates (they represent single tumours). In (**d**), data points are experimental replicates. In (**b**), one out of two independent experiments is presented. In (**a, c, d, f, g**), data are shown as mean ± SD whereas in (**e**), data are shown as mean ± SEM. In (**a, c**, and in **e–g**), *p* values were calculated by two-tailed Student's *t* test. In (**d**), *p* values were calculated by one-tailed Student's *t* test. Source data are provided as a Source Data file.

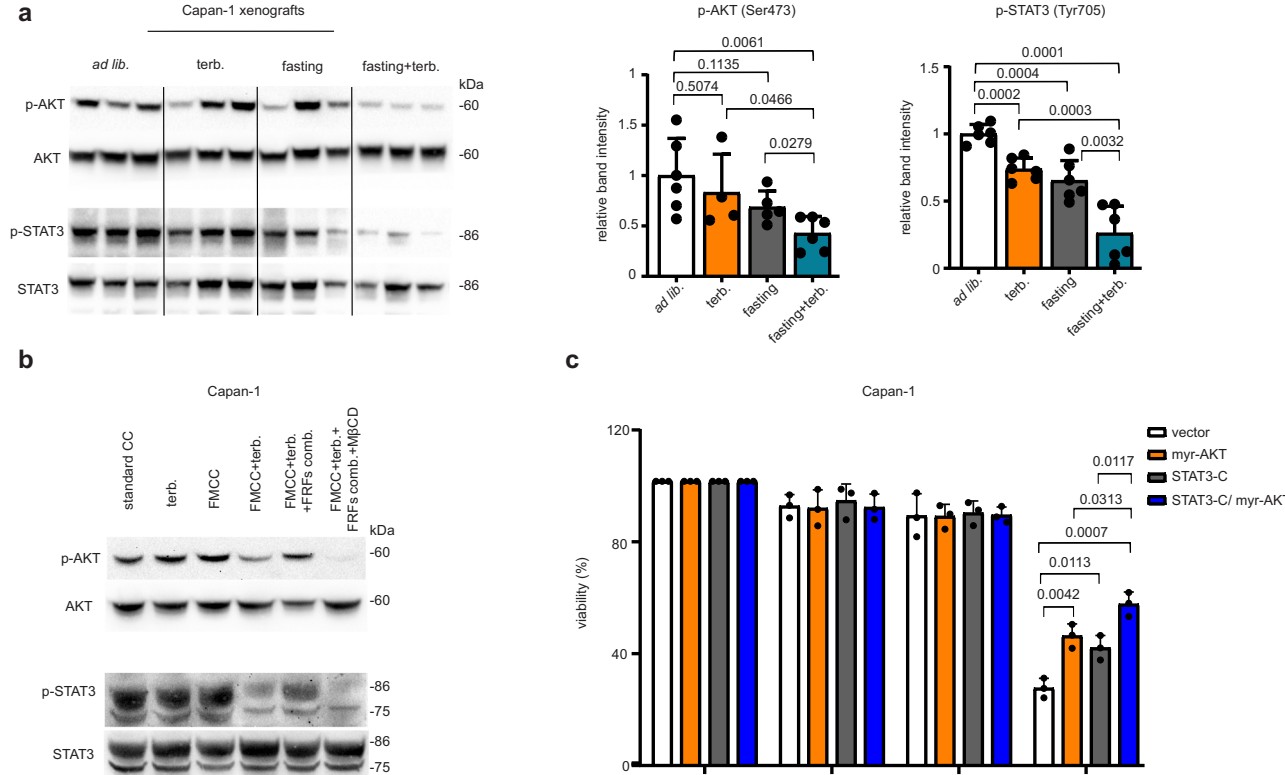

**Fig. 6 | Fasting and CBIs inhibit AKT and STAT3 signalling in cancer cells.**
**a** Tumours (Capan-1 xenografts) were isolated at the end of the experiment presented in Fig. 2a and utilized for protein lysate generation and for the detection of phosphorylated (Ser 473) and total AKT, phosphorylated (Tyr705) and total STAT3 by Western blotting. The intensity of the phosphorylated AKT and STAT3 bands were quantified and normalized to that of the total AKT and STAT3 bands, respectively. **b** Capan-1 cells were treated for 24 h w/ or w/o FMCC, combined IGF1 (5 ng/ml)+leptin (50 ng/ml)+insulin (500 pM) ("fasting-reduced factors" - FRFs-combination). Thereafter, 3.5 mM methyl-β-cyclodextrin (MβCD) w/ or w/o FMCC was added for 3 h where indicated. Afterwards, MβCD was replaced by 20 μM terbinafine for 24 h where indicated. Finally, cells were used for protein lysate generation and phosphorylated (Ser473) and total AKT, as well as phosphorylated (Tyr705) and total STAT3 were detected by immunoblotting. **c** Capan-1 cells were transduced with a constitutively active form of STAT3 (STAT3-C), with myr-AKT or with both STAT3-C and myr-AKT. Thereafter, cells were treated for 96 h w/ or w/o FMCC and during the last 72 h w/ or w/o 25 μM terbinafine. In the Western blot band quantifications presented in (**a**), samples derive from the same experiment; blots were processed in parallel; data points are biological replicates (they represent single tumours). In (**b**), one out of two independent experiments is presented. In (**c**), data points are experimental replicates. In (**a**, **c**), data are shown as mean ± SD. *p* values were calculated by two-tailed Student's *t* test. Source data are provided as a Source Data file.

phosphorylation in cancer cells by helping reduce intracellular cholesterol concentration. Ultimately, AKT inhibition plays an important role in the antitumour activity of combined CBIs and fasting.

In subsequent experiments, we focused on JAK-STAT signalling, using STAT3 phosphorylation as a readout. STAT3 is phosphorylated and thereby activated by receptor-associated JAK in response to stimuli such as leptin or IL6 and consequently translocates to the cell nucleus, where it promotes the expression of genes with anti-apoptotic and pro-proliferative effects[36]. CBIs and fasting cooperated to reduce STAT3 phosphorylation in both Capan-1 and HCT116 xenografts (Fig. 6a and Supplementary Fig. 7a, respectively). LDL cholesterol add-back prevented the reduction in phosphorylated STAT3 in HCT116 xenografts treated with clotrimazole plus weekly fasting, again indicating a role for cholesterol depletion achieved by combined CBIs and diet in STAT3 inhibition (Supplementary Fig. 7a). Signalling through lipid raft microdomains was proposed to be a mechanism through which extracellular clues activate STAT signalling, including STAT3[37, 38]. Accordingly, in lipid rafts isolated from HCT116 after treatment with combined FMCC and clotrimazole (which, by reducing intracellular cholesterol, conceivably disrupted membrane lipid raft integrity), STAT3 phosphorylation was abolished (Supplementary Fig. 7b). As in the case of AKT phosphorylation, we found that supplementation of the cell culture medium with insulin, IGF1 and leptin prevented the inhibition of STAT3 phosphorylation by combined terbinafine and FMCC (Fig. 6b). However, this rescue of STAT3 phosphorylation by insulin, IGF1 and leptin was no longer detectable when the cancer cells were treated with the cholesterol-depleting agent MβCD. Therefore, these data indicate that insulin, IGF1 and leptin restore STAT3 activation in cancer cells treated with FMCC plus CBIs through their ability to increase intracellular cholesterol content (Fig. 3d). Engineering HCT116 and Capan-1 cells to express a constitutively active STAT3 (STAT3-C)[39] conferred a partial protection from the antitumour activity of combined FMCC and terbinafine or FMCC and clotrimazole in HCT116 and in Capan-1 cells (Supplementary Fig. 7c, d). Transducing Capan-1 cells with both STAT3-C and myr-AKT conferred higher protection against combined CBIs and FMCC compared to transduction with just STAT3-C or myr-AKT (Fig. 6c). These results indicated that both AKT and STAT3 inhibition through reduced cholesterol play an important role in the antitumour effects of combined CBIs and fasting.

Coupling fasting and CBIs increased ERK, JNK and p38 phosphorylation in Capan-1 xenografts (Supplementary Fig. 8a). One possible explanation for this effect is the oxidative stress that was experienced by the tumour xenografts in response to the combined treatments, MAPK signalling being part of the cell response to reactive oxygen species[40]. Indeed, we detected increased levels of malondialdehyde (a product of peroxidation of polyunsaturated fatty acids) and increased activity of enzymes that counter oxidative stress (i.e., catalase, glucose-6-phosphate dehydrogenase and glutathione reductase) in tumours treated with CBIs and, to a higher extent, with

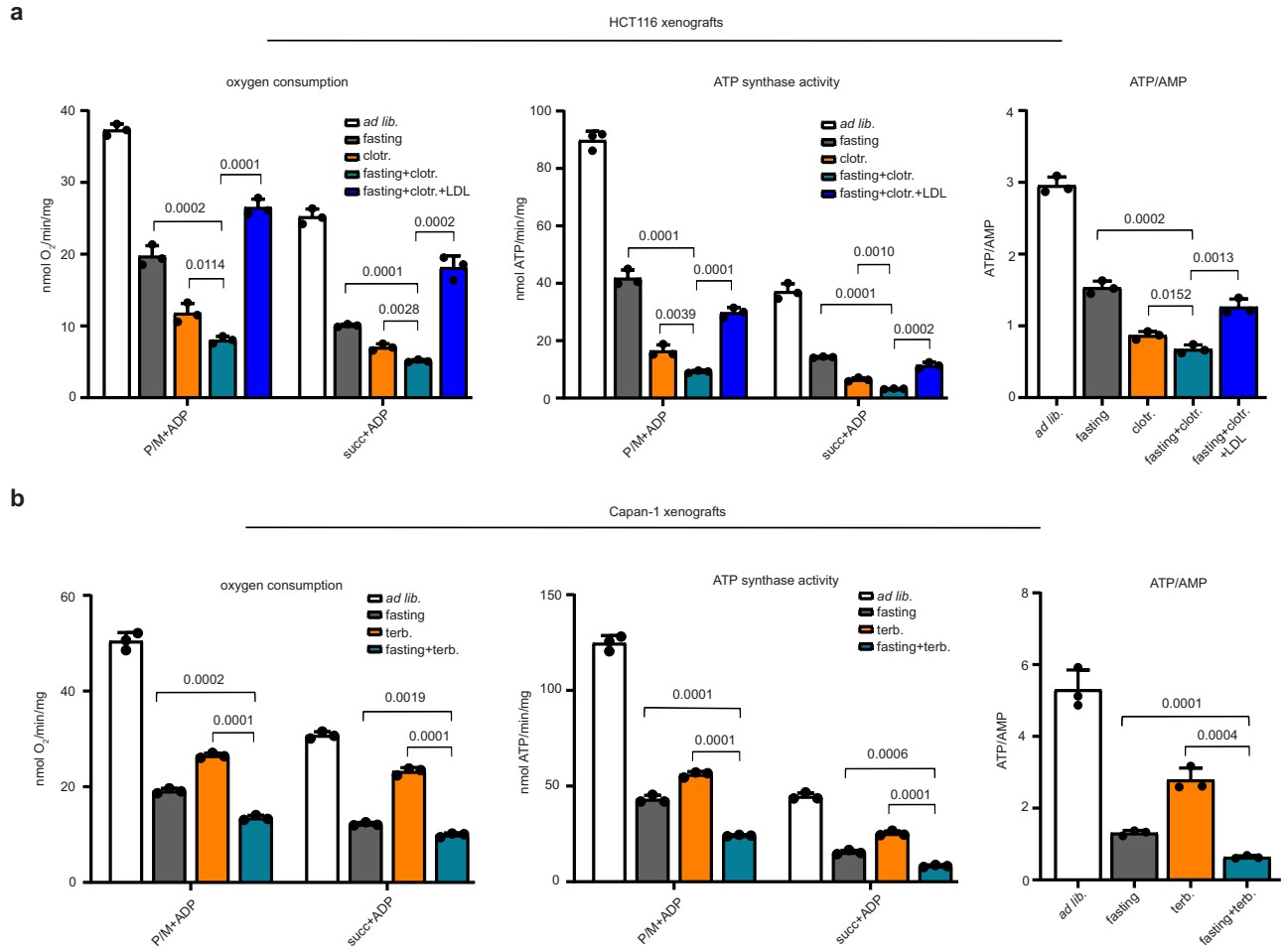

**Fig. 7 | CBIs and fasting reduce OXPHOS and energy stores in tumours.** Oxygen consumption, activity of Fo-F1 ATP synthase and ATP/AMP ratio were measured in the tumours isolated at the end of the experiment shown in Fig. 2h (HCT116 xenografts) (**a**) and Fig. 2a (Capan-1 xenografts) (**b**). Data points are biological replicates (they represent single tumours). Data are shown as mean ± SD. *p* values were calculated by two-tailed Student's *t* test. Source data are provided as a Source Data file.

CBIs plus fasting (Supplementary Fig. 8b). JNK and p38, as well as excessive ROS production were reported to exert antiproliferative or even cytotoxic effects in certain contexts[41]. However, when testing whether a JNK inhibitor (SP600125), a p38 inhibitor (SB203580) or the antioxidant, N-acetylcysteine (NAC), would affect the antitumour activity of combined CBIs and FMCC, we failed to detect an effect of these inhibitors (Suplementary Fig. 8c, d). In addition, MAPK signalling activation was not consistently observed in response to combined fasting and CBIs in all of our models, e.g. no increase in MAPK signalling was found in HCT116 xenografts (Supplementary Fig. 9a) treated with CBIs, fasting or their combination (even though increased oxidative stress markers were also present in this model, Supplementary Fig. 9b). Therefore, we did not further investigate the role of MAPK signalling activation in the cooperation between fasting and CBIs.

**Fasting plus CBIs reduce tumour oxidative phosphorylation**
Studies show that mitochondrial oxidative phosphorylation (OXPHOS) is active in cancer cells or even more in the latter than in healthy tissues and that it plays an important role in cancer progression, survival, and metastasis[42,43]. Cholesterol depletion through MβCD or statins was shown to reduce OXPHOS and cellular energy supplies via disruption of mitochondrial raft-like microdomains[44,45]. In addition, both inhibition of STAT3 and of AKT, as observed through the combination of CBIs with fasting, were also expected to hamper OXPHOS[46,47]. Thus, we investigated whether CBIs, fasting and their combination would

indeed affect OXPHOS and energy stores in HCT116 and in Capan-1 tumour xenografts by monitoring oxygen consumption, ATP synthase activity and ATP/AMP ratio. Both fasting and CBIs reduced oxygen consumption and ATP synthesis in tumours. This effect was achieved through impaired activity of both Complex I (pyruvate/malate-dependent)- and Complex II (succinate-dependent)-driven pathways (Fig. 7a, b). Combining the two interventions compounded OXPHOS obstruction. In HCT116 xenografts, LDL cholesterol add-back prevented OXPHOS inhibition achieved by the combined treatments (Fig. 7a). Therefore, fasting and CBIs cooperate to reduce OXPHOS in gastrointestinal tumour xenografts by lowering cholesterol availability.

## Discussion
Here, by performing screenings of clinically approved drugs and bioactive compounds, we identified several agents targeting different enzymes from the de novo cholesterol biosynthesis pathway, including simvastatin, which have their cholesterol-lowering effects and anticancer properties strongly enhanced through fasting. Several studies demonstrate the pro-oncogenic effects of cholesterol and of cholesterol biosynthetic enzymes[24]. Enzymes involved in de novo cholesterol production are commonly deregulated in cancer cells[24,48]. PDAC cells were shown to acquire cholesterol from their extracellular environment, but also to maintain cholesterol homoeostasis through de novo synthesis when extracellular availability is limited[49].

Alterations in free sterol levels, including free cholesterol, were detected in PDAC models and in clinical specimens of PDAC patients, and elevated expression of several genes from the cholesterol synthesis pathway, including SQLE, was demonstrated to confer a worse prognosis to these patients[50]. Similarly, cholesterol was shown to have a central role in CRC progression[51]. Mechanistically, cholesterol is required by rapidly proliferating cells for the synthesis of new membranes[24]. In addition, cholesterol accumulates in specific membrane microdomains termed lipid rafts and stabilizes them by interacting with other lipids and with specific membrane-associated proteins[24,52]. In turn, lipid rafts harbour signalling proteins with important pro-proliferative, anti-apoptotic and growth-promoting effects, including AKT and STAT3[35,37,38]. Therefore, our findings that fasting enhances CBIs' ability to lower intratumour cholesterol and to exert antitumour effects are of considerable interest, particularly in the light of evidence supporting the clinical feasibility and safety of fasting and modified fasting regimens[10–13].

Our data indicate that fasting enhances the ability of CBIs to lower cholesterol in cancer cells and that this effect is dependent on fasting-mediated reduction in insulin, IGF1 and leptin. We identified two mechanisms through which the fasting-induced reduction in insulin, IGF1 and leptin cooperates with CBIs to lower intratumour cholesterol and to slow tumour growth. On the one hand, consistent with studies showing that cellular signalling from growth factors increases the expression of cholesterol biosynthetic enzymes, (e.g. by enhancing SREBP2 expression and/or activity)[24], we found fasting alone or combined with a CBI to reduce the expression of several enzymes from the cholesterol biosynthetic route and the downregulation of at least some of these enzymes to be abrogated by concomitant insulin, IGF1 and leptin administration. We also demonstrated fasting to upregulate the cholesterol trasporter ABCG1 in tumours and to also downregulate ACAT1 in the HCT116 xenografts. Consistent with these results, we showed FMCC to boost cholesterol efflux from cultured HCT116 and MCF7 cells, an effect that was counteracted by the add-back of insulin, IGF1 and leptin. Thus, fasting appears to cooperate with pharmacological CBIs to lower cholesterol in cancer cells by at least two mechanisms: by downregulating cholesterol producing enzymes and by promoting cholesterol efflux.

CBIs and fasting also synergized to inhibit AKT and STAT3 in PDAC and in CRC cells, an effect that was mediated by cholesterol shortage and, likely, by lipid-rafts disruption as suggested by the strong reduction in phosphorylated AKT and STAT3 at the level these membrane domains[35,53]. In addition, the very low levels of leptin, one of the main activators of STAT3 signalling[36], that we detected in the serum of mice subjected to combined CBIs and fasting may also have contributed to impair STAT3 activation. AKT and STAT3 signalling play a key role in cancer cell growth[34]. Thus, the evidence we provide that combining CBIs with fasting cycles achieves a major inhibition of their activity strengthens the rationale for using this type of combined treatment.

CBIs, fasting and especially their combination also had profound metabolic effects in cancer cells as they impaired OXPHOS and blunted cellular energy supplies. An increasing body of evidence demonstrates that TCA cycle and OXPHOS are highly active in many types of cancer, including gastrointestinal cancers, and play an important role in cancer progression[54–56]. For instance, transcriptomic and metabolic analyses of PDAC cells revealed the reliance of the latter on mitochondrial respiration for cell survival[57], while the mitochondrial protein, HSP60, which is responsible for maintaining mitochondrial function, was shown to play a fundamental role in the growth of PDAC cells[58]. Thus the observation that CBIs given together with fasting cycles affect OXPHOS in tumours is also of relevance for cancer prevention or treatment. Blunted OXPHOS through fasting could be explained by the low availability of respiratory substrates to mitochondria (e.g., glucose) or by reduced STAT3 or AKT activity, given the ability of these proteins to enhance oxidative metabolism[46,47]. Concerning the effects of CBIs (when given alone or in combination with fasting cycles), our data are in line with studies showing that cholesterol depletion damages OXPHOS by compromising the integrity of mitochondrial raft-like microdomains[44,45]. In addition, OXPHOS inhibition could also reflect the marked inhibition of AKT and/or of STAT3 that was observed in response to the combined treatments[46,47].

As for the toxicity of these treatments, fasting alone or in combination with a CBI was found to increase circulating transaminases (a sign of liver damage) and CK (a marker of muscle injury/miopathy), while lowering circulating neutrophils and lymphocytes. In healthy, human PBMCs and in non-cancerous pancreatic ductal epithelial cells (HPNE), FMCC did not interact in a synergistic, but rather in an additive or even infra-additive fashion with CBIs, suggesting that these cells should be less sensitive than cancer cells to combined fasting and CBIs. The latter findings are also consistent with previous studies showing that in non-malignant cells, fasting may induce protection from, instead of sensitization to, chemotherapeutics and pro-oxidative agents[22]. In line with these data, adding a CBI to fasting failed to worsen the side effects that fasting by itself typically causes in mice (e.g., significant weight loss). Therefore, overall, combined fasting and CBIs appeared to be a relatively safe and well-tolerated treatment in our models. Yet, clinical trials exploring combined fasting and CBIs, should take possible liver and muscle toxicity into account.

In conclusion, our study lays the background for future clinical studies addressing safety, feasibility and efficacy of CBIs in combination with short-term fasting in patients with cancer. In addition, further screenings of clinically approved agents could reveal more drugs that lend themselves for repositioning in oncology when combined with fasting-based dietary regimens.

## Methods
### Cell lines and reagents
Capan-1 (catalogue number HTB-79), MIA PaCa-2 (catalogue number CRL-1420), PANC-1 (catalogue number CRL-1469), BxPC-3 (catalogue number CRL-1687), HCT116 (catalogue number CCL-247), HT29 (catalogue number HTB-38, CT26 (catalogue number CRL-2638)), N87 (catalogue number CRL-5822), PC3 (catalogue number CRL-1435), MCF7 (catalogue number HTB-22), MDA-MB-231 (catalogue number HTB-26), SKBR3 (catalogue number HTB-30), 4T1 (catalogue number CRL-2539), H1975 (catalogue number CRL-5908), B16 (catalogue number CRL-6322), HPNE (catalogue number CRL-4023) and A549 (catalogue number CCL-185) cell lines were purchased from the ATCC (LGC Standards S.r.l., Milan, Italy). ID8 cells (mouse ovarian cancer; catalogue number SCC145) were purchased from Sigma Aldrich S.r.l. (Italy). PK9 PDAC cells[59] were kindly provided by Dr. Georg Feldmann (University Hospital of Bonn, Bonn, Germany). OVCAR5 and OVCAR8 ovarian cancer cells (from the Developmental Therapeutics Program of the National Cancer Institute, Bethesda, MD, USA) were a kind gift of Dr. Gabriele Zoppoli (Department of Internal Medicine and Medical Specialties, University of Genoa). Cells were authenticated by DNA fingerprinting and isozyme detection. Cells were passaged for less than 6 months before their resuscitation for this study. All of our cell lines were routinely tested for mycoplasma contamination by MycoAlertTM Mycoplasma Detection Kit (Lonza, Italy). RPMI1640 and DMEM medium, FBS, penicillin and streptomycin were purchased from Thermo Fisher Scientific, Italy. Recombinant human IGF1 and recombinant human leptin were purchased from Peprotech, Italy. Insulin (Humulin R) was obtained from the Pharmacy of the IRCCS Ospedale Policlinico San Martino, Genoa, Italy. Puromycin, cholesterol-water soluble (cholesterol), methyl-β-cyclodextrin (MβCD), protease/phosphatase inhibitor cocktail, sulforhodamine B, N-acetylcysteine, oxiconazole, miconazole nitrate salt (miconazole), and clotrimazole were purchased from

Sigma Aldrich S.r.l. Italy. Terbinafine hydrochloride (terbinafine) was purchased from Abcam, Italy. Simvastatin was purchased from Targetmol, Italy. Itraconazole, butenafine HCl, liranaftate, SP600125 (JNK inhibitor) and SB203580 (p38 MAPK inhibitor) were purchased from Selleck Chemicals, Italy. LDLs and HDLs from human plasma were purchased from Thermo Fisher Scientific, Italy (#L3486) and from Sigma Aldrich S.r.l. Italy (#437641), respectively.

## Drug library screening

PK9 ($2.4 \times 10^3$ cells) or A549 ($8 \times 10^2$ cells) were plated in 96-well plates in their regular medium. Twenty-four hours later, the cell medium was removed and cells were incubated either in DMEM with 10% FBS, 1 g/L glucose (standard culture conditions, standard CC) or in DMEM with 1% FBS, 0.5 g/L glucose (FMCC). The day after, cells were treated with the compounds from the Microsource Spectrum Collection (Microsource Discovery Systems, Inc. Gaylordsville, USA) or the Selleck Preclinical/Clinical Compound Library (Selleck Chemicals LLC, Houston, TX, USA) at a final concentration of 10 μM. Viability was determined 72 h later by CellTiter96 Aqueous One assay (Promega, Italy) according to the manufacturer's instructions.

## Cell viability assays

Cancer cells were plated in 96-well plates in their regular medium. After 24 h, the medium was removed and cells were incubated in either standard CC or FMCC. Where indicated, cells were supplemented with IGF1 (5 ng/ml), leptin (50 ng/ml), insulin (500 pM), MβCD (3.5 mM; 3 h incubation), cholesterol (5 μg/ml), SP600125 (10 μM), or SB203580 (20 μM), HDLs [20 μM or 250 μM in those experiments in which final cholesterol concentration (LDL + HDL) in cell culture medium was 90 μM or 1 mM, respectively], LDLs [60 μM or 750 μM in those experiments in which final cholesterol concentration (LDL + HDL) in cell culture medium was 90 μM or 1 mM, respectively]. After 24 hs, cells were treated with or without miconazole, clotrimazole, oxiconazole, itraconazole, simvastatin, terbinafine, butenafine HCl, or liranaftate at the indicated concentrations. Viability was determined 72 h later by CellTiter 96 Aqueous One assay (Promega) according to the manufacturer's instructions.

PBMCs were isolated from buffy coats obtained from healthy donors at the Blood Centre of the Ospedale Policlinico San Martino IRCCS (Genoa, Italy) by density centrifugation (Ficoll-Paque). PBMCs were washed twice in PBS, counted and $10^6$ cells/well were incubated in 48-well plates for 72 hs w/ or w/o 2 μg/ml PHA, FMCC, simvastatin (10-30 μM), 10 μM terbinafine or 5 μM clotrimazole. Finally, PBMCs were harvested by gently pipetting and apoptosis was quantified by Annexin-V-FITC (Thermofisher Scientific) and propidium iodide staining and subsequent flow cytometry (using a Becton Dickinson FACS Calibur). We acquired 10,000 events per treatment condition and analyzed the data with the CellQuest software. PBMCs activation in response to PHA was confirmed by visually confirming the increase in cell size and the formation of cellular clusters and by evaluating the expression of HLA-DR and CD25 by flow cytometry[60].

## Organoid culture and viability assays

Tripsinized CRC-derived organoids, OMCR15-045TK and OMCR16-005TK[21], were included in single 3 μl Geltrex (Gibco-Thermo) drops, in the centre of each well, and allowed to recover in complete medium for 48 h and treated with or without clotrimazole and terbinafine in standard CC or FMCC (diluted culture medium 1:10 in DMEM/F12 without B27 and EGF) at the indicated concentrations for 192 h. Ten wells for each condition were live-monitored after treatment onset by the live-imager JuLI-Stage (Nano-Entek, Waltham, MA, USA) for 8 days. The mean area of organoids was calculated from each image by Image-J. Tumour organoid size at each time points was normalized against time 0, to compensate for variability in plating efficiency.

## Retroviral and lentiviral transduction

pBABE-puro (PBP), PBP-myr-AKT, FUGW and EF.STAT3C.Ubc.GFP were purchased from Addgene (Cambridge, MA, USA) and utilized for retro- and lentiviral transductions[10]. FUGW and EF.STAT3C.Ubc.GFP cells were sorted by FACSAria 2 sorter (BD Biosciences). Acquisition and analysis were performed with BD FACS Diva 8.0 (BD Biosciences).

## Immunoblotting

For protein lysate generation from cultured cells, $5 \times 10^5$ Capan-1, or $3 \times 10^5$ MIA PaCa-2 cells were plated in 100 mm Petri dishes in their regular medium. Twenty-four hours later, the medium was removed, and cells were washed with PBS and incubated in either standard CC or FMCC. After 24 h, cells were treated with or without miconazole or clotrimazole (10 μM) or terbinafine (30 μM) for a further 24 h period. For protein lysate generation from human CRC-derived organoids, OMCR15-045TK cells were plated in 6 well plates and allowed to grow in their complete regular medium for one week and then treated with or without clotrimazole (15 μM) and terbinafine (20 μM) in standard CC or FMCC for 24 h. Thereafter, cells and organoids were washed and lysed in lysis buffer [50 mm Tris-HCl (pH 7.5), 150 mm NaCl, 1% Nonidet P-40, and protease inhibitor mixture][10]. Protein concentration was determined according to standard Bradford assays. Protein lysates from primary tumours were obtained using TissueRuptor (Qiagen) according to the instructions of the manufacturer[10]. Thirty-five μg of proteins (for protein lysate obtained from in vitro cell culture) or 20 μg of proteins (for protein lysate obtained from tumour xenografts and CRC-derived organoids) were separated by SDS–PAGE and transferred to a PVDF membrane (Immobilon-P, Millipore S.p.A.). The following antibodies antibodies were utilized to detect specific proteins or their modifications: anti-phospho-AKT (Ser473; #4058; dilution: 1:1000), anti-AKT (#9272; dilution: 1:1000), anti-phospho SAPK/JNK (Thr183/Tyr185; #9255; dilution: 1:2000), anti-SAPK/JNK (#9252; dilution: 1:1000), anti-phospho-p44/42 MAPK (Erk1/2) (Thr202/Tyr204; #4377; dilution: 1:1000), anti-Erk1/2 (#9102; dilution: 1:1000), anti-phospho-p38 MAPK (Thr180/Tyr182; #4631; dilution: 1:1000), anti-p38 MAPK (# 8690; dilution: 1:1000), anti-SQLE (#40659; dilution: 1:1000), anti-DHCR24 (#2033; dilution: 1:1000), anti-ABCA1 (#96292; dilution: 1:1000) and anti-caveolin-1 (#3267; dilution: 1:1000) from Cell Signalling Technology, Italy; anti-phospho STAT3 (Tyr705; sc-8059; dilution: 1:200), anti-STAT3 (sc-482; dilution: 1:200), anti-β-actin (sc-47778; dilution: 1:10000), anti-Rabbit IgG-HRP (sc-2357; dilution: 1:5000) and anti-Mouse IgGκ BP-HRP (sc-516102; dilution: 1:5000) from Santa Cruz Biotechnology, Italy; anti-Cyclophilin B (#PA1-027A; dilution: 1:1000) from Invitrogen (Italy); anti-ABCG1 (#13578-1-AP; dilution: 1:1000); anti-LDLR (#10785-1-AP; dilution: 1:1000) from Proteintech, Italy. Band intensities were quantified with Quantity One SW software (Bio-Rad Laboratories, Inc.) using standard enhanced chemiluminescence.

## Quantitative real-time PCR (QPCR)

Total RNA (2 μg) from each sample was reverse transcribed to complementary DNA (cDNA) in a final reaction volume of 50 μl using the High-Capacity cDNA Reverse Transcription Kit (Cat. #4368814, Applied Biosystems, CA, USA). At the end of the reaction, 150 μl of DNase/RNase free water were added to have a final volume of 200 μl. The reaction was carried out by incubating for 10 min at 25 °C, followed by 120 min at 37 °C, 5 min at 85 °C and held at 4 °C in a Mastercycler nexus GSX1 (Eppendorf, Hamburg, Germany). The QPCR reaction was conducted in 10 μl of a reaction mixture composed of GoTaq® qPCR Master Mix (Cat. #A6002, Promega, Madison, Wi, USA), 0.3 μM of each primer and DNase/RNase free water in the Real-Time PCR System QuantStudio 5 (Applied Biosystems, CA, USA). The reaction mixture was subjected to an initial polymerase activation at 95 °C for 2 min, followed by 40 cycles of denaturation at 95 °C (15 s) and annealing and extension at 60 °C (1 min). After these 40 cycles, a melting curve (95 °C for 15 s, 60 °C for 15 s and 95 °C for 15 s) was

generated to confirm the single product. Gene-specific primers for QPCR were purchased from Sigma-Aldrich or Thermo Fisher and are listed in Supplementary Table 4. Comparisons in gene expression were performed using the $2^{-\Delta\Delta Ct}$ method[10]. HPRT1 was used as a house-keeping gene in the QPCRs that were performed in HCT116 xenografts, while ACTB was used as a housekeeping gene in the QPCRs that were done in Capan-1 xenografts.

### Lipid raft isolation
Lipid raft isolation was performed using the Minute Plasma Membrane-Derived Lipid Raft Isolation Kit (LR-042, Invent Biotechnologies). Briefly, HCT116 cells ($40 \times 10^6$) were harvested in T175 flasks in their regular medium. Twenty-four hours later, the medium was removed, and cells were washed with PBS and incubated in either standard CC or FMCC. After 24 h, cells were treated with or without clotrimazole (15 µM) for a further 24 h. Thereafter, cells were washed with ice cold PBS, and the assay was performed based on the manufacturer's instructions.

### Colony formation assay
Capan-1 cells ($1 \times 10^3$) were plated in 6-well plates in a regular medium. Twenty-four hours later, the cell medium was removed. Cells were washed twice with PBS and were incubated in either standard CC or FMCC. The next day, cells were treated w/ or w/o 15 µM clotrimazole, or 30 µM terbinafine for 24 h. Then, the cell medium was removed and cells were cultured for two additional weeks. Cell colonies were finally counted[10].

### Mice
All mouse experiments were performed in accordance with the laws and institutional guidelines for animal care and use established in the Principles of Laboratory Animal Care (directive 86 /609 /EEC). Animal work was only started upon approval by the Italian Istituto Superiore di Sanità (ISS; authorization #280/2022, protocol 22418.169). Six-eight-week-old female athymic nude mice or C57BL/6 mice (purchased from Envigo) were used in the experiments at the Animal Facility of the IRCCS Ospedale Policlinico San Martino. Animals were maintained in air-filtered laminar flow cabinets at $22 \pm 2\,°C$ and approximately 50% relative humidity under a 12-h light/dark cycle. The standard diet was purchased from Mucedola (Italy; cat.#4RF18). Mice were acclimatized for 1 week. Two million Capan-1 or HCT116 cells or $1.5 \times 10^5$ mouse B16 melanoma cells were injected subcutaneously into each flanks of the mouse. Treatment was initiated when the tumours appeared as established palpable masses (~1/2 weeks after cell injection). In each experiment, mice were randomly assigned to receive one of the following treatments or their combinations: control (*ad libitum* diet); clotrimazole (60 mg/kg twice a week in peanut oil, intraperitoneal (i.p) injections[61]); terbinafine (40 mg/kg/day - Monday-through-Friday - in 5% absolute ethanol + 95% of 0.5% methylcellulose by oral gavage[17]); simvastatin (80 mg/kg/day - Monday-through-Friday - in 0.5% methyl-cellulose, by oral gavage); IGF1 (200 ug/kg body weight, i.p., twice a day on the days of fasting); insulin (20 mU/kg body weight, i.p., on the days of fasting); leptin (1 mg/kg body weight, i.p, once a day Monday through Friday, including the days of fasting); LDL (0.25 mg/mouse[62,63], i.p., twice a week on the days of fasting, for a total of seven times); HDL (1 mg/mouse, i.p., twice a week on the days of fasting, six times); fasting (water-only, for 48 h every week); oxaliplatin (10 mg/kg, i.p., once a week in concomitance with the second day of fasting). During the 48 h of fasting, mice were housed in a clean, new cage to reduce copro-phagy and the intake of the residual chow. Body weight was measured immediately before, during, and after fasting. Fasting cycles were repeated every 7 days to allow for complete recovery of body weight before a new cycle. The size of the tumours was measured twice a week and tumour volume was calculated using the formula: tumour volume $= (w^2 \times W) \times \pi/6$, where "$w$" and "$W$" are "minor side" and "major side"

(in mm), respectively. The maximal tumour volume that was permitted by our Institutional Animal Care and Use Committee (IACUC) was 1,500 mm³, and in none of the experiments were these limits exceeded. Tumour masses were always isolated at the end of the last fasting cycle, weighed, divided into two parts, snap-frozen in liquid nitrogen and stored at −80 °C for subsequent protein and RNA extraction. Sample size estimation was performed using PS (Power and Sample size cal-culation) software (Vanderbilt University, TN, USA). By this approach, we estimated that the number of mice that were assigned to each treatment group would reach a power of 0.85. The Type I error probability associated with our tests of the null hypothesis was 0.05. Mice were assigned to the different experimental groups in a random fashion. Operators were unblinded, blinding during animal experi-ments was not possible because mice were subject to a specific diet supply and daily treatment.

### Blood counts, blood biochemistry and ELISAs in mouse serum
Mice whole blood was collected before treatment onset and at the end of the experiment in Eppendorf tubes. It was allowed to coagulate for 2 h at room temperature, centrifuged for 20 min at $2000 \times g$ and then serum was separated and stored in aliquots at −80 °C until subsequent use. Mouse blood counts and measurement of serum biochemical parameters, including total cholesterol, LDL and HDL cholesterol were performed at San Raffaele Hospital (Milan, Italy) using certified kits on the Ilab-650 automatic instrument (Werfen, Italy). Each analysis was preceded by the measurement of certified internal quality controls purchased directly from the manufacturer. ELISA kits to determine murine serum level of mouse Igf1 and leptin were purchased from R&D System, Italy, whereas the ELISA kit for mouse c-peptide was pur-chased from Alpco, Italy.

### Cholesterol quantification
Cholesterol content in cells ($1 \times 10^6$) and tumours (10 mg) was mea-sured by Cholesterol/ Cholesteryl Ester Quantification kit (ab65359, Abcam, Italy) according to the manufacturer's instructions. Choles-terol concentration was always normalized to the protein content of the respective sample (which was determined by standard Bradford method).

### Cholesterol efflux measurement
Cholesterol efflux from HCT116 and MCF7 cells was evaluated with a radioisotopic technique[64,65]. Briefly, cells were plated in 96-well plates in regular medium and were allowed to adhere. Twenty-four hours later culture medium was removed and cells were labelled with 2 µCi/ml [1,2-3H]-cholesterol (Perkin Elmer, Waltham, MA, USA) for additional 24 h in the presence of an acyl-CoA:cholesterol acyl-transferase ACAT inhibitor compound (Sandoz; Merck, Darmstadt, Germany) at a concentration of 2 µg/mL to ensure all cholesterol was in free form. Radiolabelling was carried out in standard CC w/ or w/o 20 µM terbinafine or 10 µM clotrimazole, or in FMCC w/ or w/o IGF1 (5 ng/ml)+leptin (50 ng/ml)+insulin (500 pM). Cells were then incubated for 2 h with DMEM supplemented with 0.2% fatty acid free BSA and the ACAT inhibitor [2 µg/ml]. Afterwards, the choles-terol acceptors apolipoprotein A-I [10 µg/ml] or HDL [12.5 µg/ml] were added for 4 h before the medium was collected and the radioactivity was quantified. A set of cell monolayers were washed with PBS before the incubation with cholesterol acceptors, in order to evaluate the amount of cholesterol inside the cells before the efflux phase. Cholesterol efflux was calculated as the percentage of radioactivity released into the medium over the total radioactivity incorporated by the cells.

### Oxidative phosphorylation measurement
Tumour masses were washed in PBS and homogenized by a Potter−Elvehjem system in 1 mL of PBS plus protease inhibitors. All

operations were performed on ice. Total proteins were estimated by the Bradford method[66] and used to normalize oxygen consumption and ATP synthase activity. Oxygen consumption measurements were conducted using an amperometric electrode (Unisense Micro-respiration, Denmark) while Fo-F1 ATP synthase activity was determined using a luminometer (Glomax 20/20, Luminometer, Promega, USA). The assays were performed in presence of pyruvate + malate (P/M) or succinate, to activate the pathways triggered by Complex I or Complex II, respectively[67]. Intracellular ATP and AMP concentrations were evaluated spectrophotometrically following NADP reduction or NADH oxidation. The ATP/AMP value was calculated as the ratio between the intracellular concentration of ATP and AMP, expressed in mM.

### Lipid peroxidation and antioxidants assays
Lipid peroxidation was evaluated through measurements of malondialdehyde (MDA) by the thiobarbituric acid reactive substances assay[68]. The enzymatic activity of endogenous antioxidant defenses markers catalase, glucose 6-phosphate dehydrogenase (G6PD) and glutathione reductase (GR) were measured spectrophotometrically following the decomposition of $H_2O_2$ (catalase)[69], the NADP reduction (G6PD)[70], or the oxidation of NADPH (GR)[69].

### Statistical analysis
All statistical analyses were performed using GraphPad Prism software version 8.0 (San Diego, CA, USA). The cooperative index (CI) was calculated as the sum of the degree of cell death induced by either a drug or FMCC alone divided by the cell death caused by combined drug plus FMCC. Typically, CI values < 1, = 1 and > 1 indicate a synergistic, additive or infra-additive effect, respectively. Here, for increased stringency, we selected a CI ≤ 0.8 to identify drugs that synergistically interacted with FMCC. Statistical comparisons were performed using the tests indicated in the figure legends. A $P$ value lower than 0.05 was considered to be statistically significant.

### Reporting summary
Further information on research design is available in the Nature Portfolio Reporting Summary linked to this article.

## Data availability
All data supporting the conclusions of this study can be found in the Article, Supplementary and Source Data files. Source data are provided with this paper.

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

## Acknowledgements

This work was supported in part by the Associazione Italiana per la Ricerca sul Cancro (AIRC; #22098 to A.N. and MFAG#26482 to I.C.), the Fondazione Umberto Veronesi (to I.C.), the Italian Ministry of Health (PE-2016-02362694, PE-2016-02363073), the 5 × 1000 2014 Funds to the Ospedale Policlinico San Martino IRCCS (to A.N.). The authors are thankful to Prof. Valerio Vellone (Department of Surgical Sciences,

University of Genoa) for his technical support and to Prof. Luca Liberale (Department of Internal Medicine and Medical Specialties, University of Genoa) for the helpful discussion.

## Author contributions

A.K., I.C. and A.N. conceived and designed the study. A.K., A.G. and I.C. performed most of the experiments. S.R., N.B., M.P.A., B.P., L.R., R.B., P.B., A.Namatalla., D.V., and D.R. performed in vitro experiments. F.M., M.C., L.P. and F.B. supervised in vitro experiments (F.B.) and/or contributed to write and to edit the manuscript (F.M., M.C., F.B. and L.P.). A.K., and A.N. wrote the manuscript with input from all authors. All of the authors approved the final manuscript.

## Competing interests

A.N. and I.C. hold intellectual property rights on clinical uses of modified fasting regimens. Relevant patents by A.N. and I.C. include WO2017140641A1, MI2014A000537, WO2018138090A1, USCLN0306-PUSA. The remaining authors declare no competing interests.
