## [Peer Review File · Nature Communications]

Reviewers' Comments:

Reviewer #1:

Remarks to the Author:

This study identified that fasting mimicking diet plus cholesterol biosynthesis inhibitors as a potential synergistic approach for suppressing cancer growth. While it is of potential translational value, its impact is limited by the use of simple, subcutaneous *in vivo* models that cannot fully reproduce the tumor microenvironment *in vivo*. Moreover, some key mechanistic details are missing, and there is discordance between the FMD condition *in vitro* and *in mice*.

Major points:

1. There is a major disconnection between the fasting mimicking medium as compared to fasting *in vivo*. With 1% vs 10% serum, there should be a drastic difference in the cholesterol levels in the culture medium that artificially forces cancer cells to rely on endogenous cholesterol synthesis (Figure 1), which of course will lead to a lethal phenotype with cholesterol biosynthesis blockers. On the other hand, serum cholesterol levels (LDL and HDL) hardly changed in the fasting diet alone group (Figure 2C). To exclude this experimental discrepancy, the authors should replenish 1% culture medium with LDL/HDL to approximate serum cholesterol levels *in vivo*, followed by re-evaluation of synergies with different cholesterol biosynthesis inhibitors.
2. Most of the *in vivo* experiments performed in this study are based on subcutaneous xenografts, which lack important elements of the tumor microenvironment. The authors should validate their work in spontaneous models of carcinogenesis, e.g. ApcMin/+ mice, LsL-Kras mice for colorectal and lung cancers.
3. Fasting mimicking diet have been shown to potentiate the response of cancer cells to different chemotherapy and immunotherapy regimens, such that the synergies between cholesterol inhibitors and FMD is hardly unique. The authors should benchmark cholesterol biosynthesis inhibitors against conventional chemotherapy + FMD in their mouse models.
4. Systemic distribution of cholesterol is oppositely regulated by the actions of LDL and HDL cholesterol. As such, please provide the HDL-to-LDL ratios in Figure 2C. The authors should measure reverse cholesterol transport in tumors under FMD conditions, which might be a potential point of synergy with cholesterol biosynthesis inhibition. In this connection, can the administration of HDL potentiate with cholesterol biosynthesis inhibition *in vivo*?
5. The authors then tried to link the synergistic effect of FMD and cholesterol biosynthesis inhibitors to the growth factors levels in medium and serum of mice fed FMD. Although they showed that growth factors could partially restore cholesterol levels in cells treated with FMD and cholesterol biosynthesis inhibitors. What is the molecular basis of such effect? The authors should examine the expression of genes involved in cholesterol uptake and/or biosynthesis, and employ metabolic approaches to test whether growth factors function to increase cholesterol biosynthesis or uptake.
6. To prove that growth factors-driven cholesterol is important for the activation of AKT and STAT3, the authors should perform cholesterol depletion in cells treated with growth factors.
7. The authors should perform rescue experiments with IGF1+leptin+insulin *in vivo* to prove their hypothesis of IGF1+leptin+insulin-cholesterol-AKT/STAT3 signalling underlying the effect of FMT plus cholesterol biosynthesis inhibitors.

Reviewer #2:

Remarks to the Author:

This paper provides compelling evidence for the cytotoxic effects of cholesterol biosynthesis inhibitors (CBI) in the context of (intermittent) fasting in diverse types of cancer cells. Drug screens identified azoles, squalene epoxidase inhibitors and statins as powerful killers of a wide variety of cancer cells, in particular if applied in "fasting mimicking" culture medium. The authors subsequently go on to show that terbinafine and clotrimazole, antifungal drugs that inhibit cholesterol synthesis, strongly inhibit the growth of pancreatic- and colon cancer cell xenografts in mice subjected to weekly periods of 48 hours of water-only fasting. Mechanistically, downregulation of circulating growth factors induced by fasting appears to reinforce the impact of CBI's on cholesterol synthesis, which most likely disrupts the formation of membranous "lipid rafts" (cholesterol is a crucial component of lipid rafts). The latter may explain the substantial

reduction of AKT/STAT3 phosphorylation as well as the significant disruption of oxidative phosphorylation and ATP production in cancer cells in response to CBI/fasting, since lipid rafts are critically important for transmembrane signal transduction as well as for the structure of mitochondrial raft-like microdomains. The paper makes a strong case for a significant role of downregulation of AKT/STAT3 signaling and reduction of cellular energy stores in the anticancer effects of CBIs and fasting.

I find this very convincing data suggesting that intermittent fasting may turn available cholesterol synthesis inhibitors into valuable tools for the treatment of cancer. The experiments are carefully designed and executed. The methods used are up-to-date and the statistical analyses are appropriate. The results of both in vitro and in vivo studies evaluating cancer cell growth in response to CBI/fasting are almost surprisingly clear and highly consistent, and the mechanistic explanation of the effects is firmly supported by the results of biochemical studies. However, the paper almost completely lacks data supporting the safety of the interventions.

Fasting is known to induce "differential stress resistance", rendering cancer cells vulnerable to various types of stress, while simultaneously protecting healthy cells against the same stressors. As the combination of fasting and the CBI's employed in the reported studies profoundly reduced the cholesterol content of cancer cells, and this appeared to be devastating for cell viability, it seems crucial for clinical applicability to understand if healthy cells are protected against similar effects of the combination. I'm not sure why the authors did not conduct (or report) experiments in healthy cells. Notably, quite compelling data suggest that statins alone can induce myopathy and perhaps even cognitive decline as a result of inhibition of respiratory chain complexes, reduction of oxidative phosphorylation and increased oxidative stress. The (sparse) safety data provided in Suppl Table 1 are not very reassuring in view of the doubling of ASAT and ALAT in (very few) clotrimazole treated animals and the apparent absence of signs in "standard clinical monitoring" (what is that?).

I was a bit surprised that the authors did not use simvastatin in their in vivo and mechanistic studies. Simvastatin is widely used as an oral cholesterol lowering drug, while clotrimazole is merely available for local application as crème or vaginal tablet (as far as I know), while systemic terbinafine therapy is known for its potential hepatotoxic effects.

Hanno Pijl

Reviewer #3:

Remarks to the Author:

In this manuscript, Khalifa and colleagues set out to identify approved drugs that will cooperate with cyclic fasting to act as anticancer agents. They screen drugs in fasting-mimicking culture conditions (FMCC) that contains 1% FBS and 0.5 g/L glucose. They identify azoles drugs as strong hits and further extend the observation to other cholesterol biosynthesis inhibitors (CBIs), such as statins and terbinafine. The synergist effects of FMCC and CBIs are strong and importantly observed in a wide range of cancer cell types, suggesting that these results have broad implications for therapy. Convincingly, the authors demonstrate that cyclic fasting (48 hours on water each week) potentiates the effects of CBIs as anticancer agents in mouse subQ xenografts. Remarkably, IP supplementation with LDL on the days of fasting blunts these anticancer effects, implicating cholesterol as a key nutrient for tumor growth under fasting conditions. This is consistent with the effects of CBIs.

The authors use cell culture and subQ xenografts to demonstrate that fasting + CBIs decreases cholesterol in tumors and cells. The effects of FMCC + CBIs are partially bypassed by addition of IGF1/insulin/leptin cocktail, all of which are known to be decreased during fasting. Convincing data is provided to suggest that fasting + CBIs affects tumor growth at least in part by decreasing Akt-STAT3 signaling and limiting mitochondrial oxidative phosphorylation. Indeed, effects of fasting + CBIs on OXPHOS are rescued by LDL supplementation in HCT116 subQ xenografts. Overall, the manuscript is clearly written and the data support the conclusions. This is an exciting study that has broad implications for the effects of fasting on tumor growth and possibly cancer therapy. A few comments below are provided that will strengthen the manuscript.

Major comments:

1. Figure 1B indicates that other commercial drugs were equally cooperative with FMCC medium. Please indicate the identity of these hits. In addition, it would be helpful to highlight the statin drugs tested in Fig. 1B by shading these dots with a different color.
2. In Figure 1E, the authors measure cellular cholesterol after 24 hours of different treatments. FMCC plus terbinafine is reported to decrease cholesterol to almost zero as compared to standard culture conditions. This is impossible. Say for example, cells double once in 24 hours. Even if no new cholesterol is made or taken up from the medium, cells should have 50% of their original cholesterol based on conservation of mass. Please provide an explanation and also note how rapidly Capan-1 cells divide in FMCC plus terbinafine.
3. As in #2 above, the same issue applies to Fig. 3D where cells were incubated for 48 hours with FMCC, which decreases cholesterol, and then another 24 hours with terbinafine. The effects of terbinafine do not make sense in this case as well.
4. Did the authors test whether LDL supplementation in mice also rescued growth of Capan-1 subQ xenografts? If so please report the result even if negative as the generalizability of this is treatment regimen is important.

Minor comments:

1. Page 2, line 39: "correspondeing" is misspelled.
2. page 13, line 233: the authors may have intended to say "readout" instead of "reading frame"

Genoa, July 19th 2023

Below please find our point-by-point reply to the concerns raised by the reviewers about the article "Cyclic fasting bolsters cholesterol biosynthesis inhibitors' anticancer activity" (NCOMMS-23-03940-T), by Amr Khalifa and colleagues

Reviewer #1 (Remarks to the Author):

Major points:

1. *There is a major disconnection between the fasting mimicking medium as compared to fasting in vivo. With 1% vs 10% serum, there should be a drastic difference in the cholesterol levels in the culture medium that artificially forces cancer cells to rely on endogenous cholesterol synthesis (Figure 1), which of course will lead to a lethal phenotype with cholesterol biosynthesis blockers. On the other hand, serum cholesterol levels (LDL and HDL) hardly changed in the fasting diet alone group (Figure 2C). To exclude this experimental discrepancy, the authors should replenish 1% culture medium with LDL/HDL to approximate serum cholesterol levels in vivo, followed by re-evaluation of synergies with different cholesterol biosynthesis inhibitors.*

We would like to thank the referee for raising this point. Indeed, "exposing cells to FMCC implies lowering FBS in the tissue culture medium by 90% (from 10% to 1%). As a result, cholesterol concentration in the medium will also be reduced to a similar extent. Therefore, we assessed whether FMCC-mediated enhancement of CBIs' activity reflected the reduction in cholesterol availability to cancer cells, a condition that does not occur *in vivo* during short-term fasting (see below) and that could artificially force cancer cells to rely on endogenous cholesterol biosynthesis. Cholesterol concentration in regular FBS is 0.9 mM (Ref. 19). Therefore, in a standard tissue culture medium, which contains 10% FBS, cholesterol concentration can be estimated to be ~90 μ M. We found that FMCC retained their ability to enhance CBIs's activity even when the FMCC medium was supplemented with LDL plus HDL cholesterol (at a 3:1 ratio) to recreate a final cholesterol concentration that was similar to that found in standard tissue culture medium (~90 μ M; Fig. 1f, Supplementary Fig. 1i). Even when we supplemented FMCC medium with LDL and HDL cholesterol (always at a 3:1 ratio) to reach a final cholesterol concentration that was similar to that from human interstitial fluids (which represent the microenvironment bodily tissues, including cancer cells, are exposed to; here, cholesterol concentration was found to be approximately 20% of the cholesterol plasma concentration, i.e. ~1 mM) (Ref. 20), we still found the cooperation between FMCC and CBIs to be readily detectable (Supplementary Fig. 1j, k). Thus, these findings indicated that FMCC-mediated enhancement of CBIs antitumour activity did not reflect reduced cholesterol availability to cancer cells through reduced FBS".

2. *Most of the in vivo experiments performed in this study are based on subcutaneous xenografts, which lack important elements of the tumor microenvironment. The authors should validate their*

work in spontaneous models of carcinogenesis, e.g. ApcMin/+ mice, LsL-Kras mice for colorectal and lung cancers.

We fully agree that subcutaneous tumor xenografts have limitations and that they lack important elements of the tumor microenvironments, including cells from the adaptive immune system. We did assess the possibility to perform an experiment with a genetic mouse model, such as ApcMin/+ or LsL-Kras mice. However, this entailed having a new animal work protocol approved by the competent authority, purchasing the animals, breeding them, genotyping the newborn mice, waiting that they reach the appropriate age and ultimately actually assign them to the different treatments to assess the antitumor effects of the latter. Altogether, we estimated that this would take between nine months and one year, which would not be compatible with the time allowed for the revision. We also enquired with collaborators of ours from Germany (Prof. Georg Feldmann, University of Bonn), who already make use of genetic tumor models, such as KPC mice (a pancreatic cancer model). However, also in this case, the local ethics committee expected them to write a new protocol and the estimated time for its approval and then to perform the experiment was similar to the one required at our institution. Yet, given the importance of the issue that was raised, we performed an experiment with immunocompetent mice (C57BL/6), which we used to establish allografts of B16 melanoma cells (a model which is approved for use at our institution), having verified that these cells are also sensitized to the antitumor effects of cholesterol inhibitors by starvation (new Supplementary Fig. 2c). In this immunocompetent mouse tumor model, a striking sensitization of the tumor to clotrimazole was also observed (new Supplementary Fig. 4c), essentially validating our approach in mice with a tumor microenvironment that more closely resembles human tumors (given the presence of functional T and B cells in these animals).

- 3. Fasting mimicking diet have been shown to potentiate the response of cancer cells to different chemotherapy and immunotherapy regimens, such that the synergies between cholesterol inhibitors and FMD is hardly unique. The authors should benchmark cholesterol biosynthesis inhibitors against conventional chemotherapy + FMD in their mouse models.*

We compared the combination of fasting plus terbinafine vs. fasting plus oxaliplatin (a chemotherapeutic that is commonly used against g.i. tumors and which was shown to become more active when combined with fasting - Bianchi et al., *Oncotarget* 2015;6:11806) in terms of their effect on the growth of Capan-1 xenografts (new Supplementary Fig. 4d). Cycles of fasting enhanced the antitumor activity of both terbinafine and oxaliplatin. The two combinations (terbinafine+fasting and oxaliplatin+fasting) exhibited a similar ability to reduce tumor growth with essentially not difference between the two regimens.

- 4. Systemic distribution of cholesterol is oppositely regulated by the actions of LDL and HDL cholesterol. As such, please provide the HDL-to-LDL ratios in Figure 2C. The authors should measure reverse cholesterol transport in tumors under FMD conditions, which might be a potential point of synergy with cholesterol biosynthesis inhibition. In this connection, can the administration of HDL potentiate with cholesterol biosynthesis inhibition in vivo?*

We are indebted with the reviewer for asking that we perform these experiments. We provide the HDL-to-LDL ratios in former Figure 2C (now Fig. 2e). We also performed several experiments to address whether fasting may cooperate with CBIs by enhancing cholesterol efflux. Indeed, as mentioned in our Results section (pages 12-14), "... not only does the intracellular cholesterol content depend on cholesterol biosynthesis and on the uptake of circulating cholesterol, but it also depends on the degree of cholesterol efflux from the cell (Ref. 27). Non-esterified cholesterol is effluxed from cells to extracellular HDLs through the action of transporters, such as ATP Binding Cassette Subfamily A Member 1 (ABCA1) and ATP Binding Cassette Subfamily G Member 1 (ABCG1) (Ref. 27). Specifically, ABCA1 mediates the secretion of cellular free cholesterol to the extracellular acceptor, apolipoprotein AI (APO-AI), to form nascent HDL, while ABCG1 promotes free cholesterol efflux to mature HDL. Studies show that insulin (Refs. 28-30), IGF1 (Refs. 31-33) and leptin (Ref. 34) all have the ability to negatively affect cholesterol efflux. In this context, insulin was shown to dampen ABCG1 expression and to down-regulate ABCA1 activity through its specific phosphorylation (Refs. 29, 30). Leptin was

reported to accelerate cholesteryl ester accumulation in human macrophages by increasing ACAT1 expression via JAK2 and PI3K, thereby suppressing HDL-mediated cholesterol efflux (Ref. 34). In Capan-1 xenografts, we found weekly fasting and fasting plus terbinafine to increase ABCG1 expression both at the mRNA and at the protein level (new Fig. 5a). However, when insulin, IGF1 and leptin were administered to the mice receiving fasting plus terbinafine, ABCG1 upregulation was abolished (new Fig. 5a). No increase in ABCA1 expression in response to these treatments was observed (new Fig. 5b and data not shown). A higher ABCG1 expression in response to fasting plus simvastatin was also detected in HCT116 xenografts treated with simvastatin plus fasting (new Fig. 5c). In the latter model (but not in Capan-1 tumours), combined fasting and terbinafine also downregulated ACAT1 expression (new Fig. 5c). Consistent with the *in vivo* evidence that FMCC stimulates ABCG1 and not ABCA1 expression, we observed that *in vitro* cholesterol efflux from both HCT116 and MCF7 cancer cells only increased to HDL and not to ApoA-I under FMCC (new Fig. 5d). Moreover, when the FMCC medium was supplemented with insulin, IGF1 and leptin, the enhancement in cholesterol efflux through FMCC was either no longer detectable (MCF7 cells) or occurred at a lower extent as compared to regular FMCC (HCT116) (new Fig. 5d). Therefore, these findings are consistent with fasting promoting cholesterol efflux in these cancer cells through reduced insulin, IGF1 and leptin levels. Finally, in line with the ability of fasting to increase ABCG1 expression in tumours *in vivo* and thus, conceivably, to enhance cholesterol efflux to mature HDLs, we found the administration of HDLs to Capan-1 xenograft-bearing mice to increase fasting's ability to reduce intratumor cholesterol and to slow tumour growth (new Fig. 5e-g). Overall, these results indicate that fasting promotes cholesterol efflux via ABCG1 and possibly also via reduced ACAT1 expression, at least in some types of tumour cells. In turn, fasting-induced cholesterol efflux is likely to contribute to lower cholesterol content inside cancer cells.

Given the limited number of animals we had available, we were not able to test the effect of combined HDL plus CBIs *in vivo*. We agree that this combination would have been very interesting to test. However, again, having few animal groups available and also based on the results we obtained (i.e. increased ABCG1 expression in tumors in response to fasting and increased cholesterol efflux in fasted cancer cells), we preferred to test whether the HDLs would enhance fasting's antitumor effects. We hope that this will be acceptable.

5. *The authors then tried to link the synergistic effect of FMD and cholesterol biosynthesis inhibitors to the growth factors levels in medium and serum of mice fed FMD. Although they showed that growth factors could partially restore cholesterol levels in cells treated with FMD and cholesterol biosynthesis inhibitors. What is the molecular basis of such effect? The authors should examine the expression of genes involved in cholesterol uptake and/or biosynthesis, and employ metabolic approaches to test whether growth factors function to increase cholesterol biosynthesis or uptake.*

Again, we would like to thank the referee for raising this point and for encouraging us to look into the mechanisms whereby low insulin, IGF1 and leptin (as a result of fasting) affect cholesterol metabolism. To address this point, "... we first focused on the expression of enzymes from the cholesterol biosynthesis pathway. This in view of studies showing that these circulating factors and the signaling cascades they trigger (such as the PI3K-AKT-mTOR pathway) can activate the cholesterol production pathway through various mechanisms, including increased expression and activity of sterol regulatory element-binding protein 2 (SREBP2), a transcription factor that promotes the expression of genes encoding for cholesterol-producing enzymes (Refs. 25,26). In Capan-1 xenografts, we found the expression of several enzymes involved in cholesterol biosynthesis, such as 3-hydroxy-3-methylglutaryl-CoA synthase 1 (HMGCS1), isopentenyl-diphosphate delta isomerase 1 (IDI-1), farnesyl diphosphate synthase (FDPS), lanosterol synthase (LSS), 24-dehydrocholesterol reductase (DHCR24) and 7-dehydrocholesterol reductase (DHCR7) to be reduced in response to fasting and/or to fasting plus terbinafine (new Fig. 4a). The add-back of insulin, IGF1 and leptin prevented HMGCS1, IDI-1, FDPS and LSS downregulation occurring in response to fasting plus terbinafine, whereas in the case of DHCR24 and DHCR7 expression, we could not detect a significant rescue effect in response to the addback of the three factors. We could not demonstrate any effect of fasting or fasting plus terbinafine on LDL receptor expression, both at the mRNA and at the protein level

(new Fig. 4b), suggesting that LDL receptor-mediated cholesterol uptake is unlikely to be affected by these interventions. We also readily detected downregulated DHCR24 and SQLE expression in protein lysates from HCT116 xenografts which were treated with weekly fasting plus clotrimazole (new Fig. 4b). Altogether, these findings indicate that several cholesterol-producing enzymes become downregulated in tumour xenografts after treatment with fasting and/or with fasting combined with a CBI and that the fasting-induced reduction in circulating insulin, IGF1 and leptin takes part in the downregulation of at least some of these enzymes”.

In addition, as mentioned above, we focused on the effects of fasting on cholesterol efflux in cancer cells and found that the reduction in circulating insulin, IGF1 and leptin that is induced by fasting promotes cholesterol efflux via ABCG1 and possibly also via reduced ACAT1 expression, at least in some types of tumour cells (see point #4). In turn, fasting-induced cholesterol efflux is likely to contribute to further lower cholesterol content inside cancer cells.

6. *To prove that growth factors-driven cholesterol is important for the activation of AKT and STAT3, the authors should perform cholesterol depletion in cells treated with growth factors.*

To address this point, we made use of the cholesterol-depleting agent methyl- β -cyclodextrin (M β CD). Restoration of AKT and STAT3 phosphorylation by insulin, IGF1 and leptin in cells treated with terbinafine plus FMCC was prevented by intracellular cholesterol depletion with M β CD (new Fig. 6b), supporting the notion that the effect of supplemented insulin, IGF1 and leptin on AKT and STAT3 activation (in cells treated with FMCC plus CBIs) reflects their ability to increase cholesterol content inside the cell.

7. *The authors should perform rescue experiments with IGF1+leptin+insulin in vivo to prove their hypothesis of IGF1+leptin+insulin-cholesterol-AKT/STAT3 signalling underlying the effect of FMT plus cholesterol biosynthesis inhibitors.*

We evaluated the effect of the add-back of insulin, IGF1 and leptin on the antitumor effect of combined terbinafine and fasting in Capan-1-xenograft bearing mice. Here, we found that terbinafine potentiation via fasting cycles was completely abolished by the add-back of these factors (new Fig. 3e). Thus, these findings confirmed the crucial role of the reduction in insulin, IGF1 and leptin in the potentiation of this CBI through fasting.

Reviewer #2 (Remarks to the Author):

1. *The paper almost completely lacks data supporting the safety of the interventions. Fasting is known to induce “differential stress resistance”, rendering cancer cells vulnerable to various types of stress, while simultaneously protecting healthy cells against the same stressors. As the combination of fasting and the CBI’s employed in the reported studies profoundly reduced the cholesterol content of cancer cells, and this appeared to be devastating for cell viability, it seems crucial for clinical applicability to understand if healthy cells are protected against similar effects of the combination. I’m not sure why the authors did not conduct (or report) experiments in healthy cells. Notably, quite compelling data suggest that statins alone can induce myopathy and perhaps even cognitive decline as a result of inhibition of respiratory chain complexes, reduction of oxidative phosphorylation and increased oxidative stress. The (sparse) safety data provided in Suppl Table 1 are not very reassuring in view of the doubling of ASAT and ALAT in (very few) clotrimazole treated animals and the apparent absence of signs in “standard clinical monitoring” (what is that?).*

We agree with the referee that the potential toxicity of these agents, particularly when coupled with fasting (which makes their cholesterol lowering effects more pronounced - at least in cancer cells) is of concern. Concerning the clinical signs of toxicity we monitored in our mouse studies, these typically included mouse behavior (i.e. reluctance to move, lethargy/apathy, persistent immobility, etc.), posture, bodily functions and weight loss. All of these parameters are routinely checked during our experiments, also because the appearance of some clinical signs (particularly of the most severe ones) may mandate performing euthanasia. However, in no case, fasting, cholesterol-lowering agents or their combination caused signs of severe distress or toxicity in our hands (see below).

For the revision of the article, we tested the possibility that fasting cycles may enhance the antitumor properties of simvastatin (as requested by Referee#2) and again used the serum collected from these mice at the end of the experiment to check for signs of organ damage. In addition, we also tested the effects of fasting-mimicking culture conditions and CBIs in cultured non-malignant cells. The results are discussed in the text (in the new Section “Toxicity of fasting, CBIs and their combination”) and are presented in New Supplementary Fig. 5, Supplementary Table 2 and new Supplementary Table 3: “Supplementary Table 2 shows the blood counts and the biochemistry tests we performed in animals receiving clotrimazole, weekly 48h fasting or their combination w/ or w/o add back of LDL-cholesterol (Fig. 2h), while Supplementary Table 3 shows the blood tests that we performed in mice treated with simvastatin, weekly fasting or their combination (Fig. 2f). In both experiments, fasting caused an increase in serum aspartate aminotransferase (AST), a marker of liver damage. Combined fasting and simvastatin raised both the levels of AST and of alanine aminotransferase (ALT), another marker of acute liver damage, while in response to fasting plus clotrimazole, an increase in AST could not be documented. In both experiments, fasting, but not fasting plus the CBI, was also associated with higher serum levels of creatine kinase (CK), a marker of muscle injury/myopathy. The fact that we could not document a significant increase in CK in response to fasting+CBIs may also have reflected the high variability of these results and the limited number of samples that we were able to analyze. Concerning the hematological toxicity of these treatments, fasting, but not fasting plus clotrimazole, reduced total white blood cells, neutrophils and lymphocytes (Supplementary Table 2). The clinical parameters that we routinely monitored to detect pain or distress in mice (i.e. behavior, posture, bodily functions, weight loss) failed to show signs of higher toxicity of fasting plus a CBI as compared to just weekly fasting. For instance, no worsening of fasting-induced weight loss could be detected with the concomitant administration of fasting plus a CBI as compared with just fasting and neither did adding a CBI to fasting affect mouse ability to re-gain weight after the fasting cycles (Supplementary Fig. 5a, b).

We also evaluated potential toxicities of fasting, CBIs and their combination *in vitro*, using non-malignant human cells, such as peripheral blood mononuclear cells (PBMCs) which mostly consist of CD3+ T lymphocytes, and the pancreatic ductal epithelial cell line, HPNE. As compared to resting/unstimulated PBMCs, PBMCs that were stimulated with the mitogen phytohemagglutinin (PHA) showed higher sensitivity to FMCC, since they became more apoptotic in response to these conditions (Supplementary Fig. 5c). However, neither in resting nor in activated PBMCs, did we detect a cooperation between FMCC and CBIs, including simvastatin, in terms of ability to induce apoptosis. HPNE cells also proved very sensitive to FMCC. However, again, neither the chemotherapeutic, oxaliplatin, nor the CBIs acted synergistically with FMCC to reduce cell viability (Supplementary Fig. 5d). Instead, the interaction between drugs and FMCC was mostly just additive or even infra-additive (antagonistic) in these cells”.

We also discussed these findings in the Discussion section of our article, where we mention that “... overall, combined fasting and CBIs appeared to be a relatively safe and well-tolerated treatment in our models. Yet, future clinical trials exploring combined fasting and CBIs, should take possible liver and muscle toxicity into account”.

2. *I was a bit surprised that the authors did not use simvastatin in their in vivo and mechanistic studies. Simvastatin is widely used as an oral cholesterol lowering drug, while clotrimazole is merely available for local application as crème or vaginal tablet (as far as I know), while systemic terbinafine therapy is known for its potential hepatotoxic effects.*

We fully agree with the referee and we thank him for raising this point. We tested whether weekly fasting cycles would enhance the ability of simvastatin to slow the growth of HCT116 colon cancer xenografts and indeed found a remarkable cooperation between the statin and the dietary intervention both in terms of *in vivo* tumour growth (new Fig. 2f, g) and of AKT inhibition (new Supplementary Fig. 6b). Thus, these findings strengthen the idea of a strong cooperation between CBIs, including statins, and periodic fasting in controlling tumor growth.

Reviewer #3 (Remarks to the Author):

Major comments:

1. *Figure 1B indicates that other commercial drugs were equally cooperative with FMCC medium. Please indicate the identity of these hits. In addition, it would be helpful to highlight the statin drugs tested in Fig. 1B by shading these dots with a different color.*

We listed the additional agents which we found to be enhanced in their antitumor activity via FMCC medium in PK9 cells in the new Supplementary Table 1. The drugs that acquired increased cytotoxic effects through FMCC in A549 lung cancer cells (Supplementary Fig. 2a) will be included in the raw data of our article. Concerning the request to highlight the statins tested in Fig. 1b, in this screen we actually did not have any statin among the drugs we tested. We started testing simvastatin afterwards in our follow-up experiments.

2. *In Figure 1E, the authors measure cellular cholesterol after 24 hours of different treatments. FMCC plus terbinafine is reported to decrease cholesterol to almost zero as compared to standard culture conditions. This is impossible. Say for example, cells double once in 24 hours. Even if no new cholesterol is made or taken up from the medium, cells should have 50% of their original cholesterol based on conservation of mass. Please provide an explanation and also note how rapidly Capan-1 cells divide in FMCC plus terbinafine.*

We would like to thank the reviewer for raising this point, which prompted us to investigate the mechanisms that could justify such a reduction in intracellular cholesterol, which, as the reviewer rightfully points out, could not be justified by just an obstruction in *de novo* cholesterol biosynthesis. Indeed, as mentioned above, "... not only does the intracellular cholesterol content depend on cholesterol biosynthesis and on the uptake of circulating cholesterol, but it also depends on the degree of cholesterol efflux from the cell (ref. 27). This is considered to be the first step of cholesterol return from peripheral tissues to the liver for biliary excretion, a process also known as reverse cholesterol transport (Ref. 27). Non-esterified cholesterol is effluxed from cells to extracellular HDLs through the action of transporters, such as ATP Binding Cassette Subfamily A Member 1 (ABCA1) and ATP Binding Cassette Subfamily G Member 1 (ABCG1) (Ref. 27). Specifically, ABCA1 mediates the secretion of cellular free cholesterol to the extracellular acceptor, apolipoprotein AI (APO-AI), to form nascent HDL, while ABCG1 promotes free cholesterol efflux to mature HDL. Studies show that insulin (Refs. 28-30), IGF1 (Refs. 31-33) and leptin (Ref. 34) all have the ability to reduce cholesterol efflux. For instance, insulin was shown to dampen ABCG1 expression and to down-regulate ABCA1 activity through its specific phosphorylation (Refs. 29, 30). Leptin was reported to accelerate cholesteryl ester accumulation in human macrophages by increasing ACAT1 expression via JAK2 and PI3K, thereby suppressing HDL-mediated cholesterol efflux (Ref. 34). In Capan-1 xenografts, we found weekly fasting and fasting plus terbinafine to increase ABCG1 expression both at the mRNA and at the protein level (new Fig. 5a). However, when insulin, IGF1 and leptin were administered to the mice receiving fasting plus terbinafine, ABCG1 upregulation was abolished (new Fig. 5a). No increase in ABCA1 expression in response to these treatments was observed (new Fig. 5b and data not shown). A higher ABCG1 expression in response to fasting plus simvastatin was also detected in HCT116 xenografts treated with simvastatin plus fasting (new Fig. 5c). In the latter model (but not in Capan-1 tumours), combined fasting and terbinafine also downregulated ACAT1 expression (new Fig. 5c). Consistent with the *in vivo* evidence that FMCC stimulates ABCG1, but not ABCA1 expression, we observed that *in vitro* cholesterol efflux from both HCT116 and MCF7 cancer cells only increased to HDL and not to ApoA-I under FMCC (new Fig. 5d). Moreover, when the FMCC medium was supplemented with insulin, IGF1 and leptin, the enhancement in cholesterol efflux through FMCC was either no longer detectable (MCF7 cells) or occurred at a lower extent as compared to regular FMCC (HCT116) (new Fig. 5d). Therefore, these findings are consistent with fasting promoting cholesterol efflux in these cancer cells through reduced insulin, IGF1 and leptin levels. Finally, in line with the ability of fasting to increase ABCG1 expression in tumours *in vivo* and thus, conceivably, to enhance cholesterol efflux to mature HDLs, we found the administration of HDLs to Capan-1 xenograft-bearing mice to increase fasting's ability to reduce intratumor cholesterol and to slow tumour growth (new Fig. 5e-g). Overall, these results indicate

that fasting promotes cholesterol efflux via ABCG1 and possibly also via reduced ACAT1 expression, at least in some types of tumour cells. In turn, fasting-induced cholesterol efflux is likely to contribute to lower cholesterol content inside cancer cells.

We believe that the increased cholesterol efflux in response to FMCC combined with a reduced expression of cholesterol biosynthetic enzymes (see new Fig. 4) justify the strong reduction in intracellular cholesterol observed with FMCC plus CBIs.

As per advice of the reviewer, we also determined how rapidly Capan-1 cell proliferate in regular culture conditions, in FMCC and upon treatment with terbinafine w/ or w/o FMCC. Upon combined treatment with FMCC plus terbinafine, cell proliferation became extremely slow. In fact, in response to this type of combined treatment, Capan-1 cells failed to double their number during our observation period (3 days; data not shown). So, based on this experiment, a “paradoxical” increase in cancer cell proliferation in response to combined FMCC plus CBIs (which could also have explained the strong reduction in intracellular cholesterol that we observed with FMCC+CBIs - due to increased cholesterol use) did not seem to occur.

3. *As in #2 above, the same issue applies to Fig. 3D where cells were incubated for 48 hours with FMCC, which decreases cholesterol, and then another 24 hours with terbinafine. The effects of terbinafine do not make sense in this case as well.*

Please see the previous point.

4. *Did the authors test whether LDL supplementation in mice also rescued growth of Capan-1 subQ xenografts? If so please report the result even if negative as the generalizability of this is treatment regimen is important.*

Previously, we had not evaluated whether LDL supplementation would also rescue the growth of Capan-1 subcutaneous xenografts in response to a cholesterol biosynthesis inhibitor plus cycles of fasting. However, we agree that confirming this rescue affect in a second model is important. So, we performed this experiment and verified that indeed the add-back of LDL cholesterol also fully abrogates the enhancement of terbinafine antitumor activity via fasting in Capan-1 xenograft-bearing mice (new Supplementary Fig. 3e).

Minor comments:

1. *Page 2, line 39: “correspondeing” is misspelled.*
Thanks for highlighting this mistake, which we have now corrected.
2. *Page 13, line 233: the authors may have intended to say “readout” instead of “reading frame”*
Indeed this was a mistake which we have now corrected.

Thanks again for your consideration.

With my warmest regards,

Alessio Nencioni, MD

Email: alessio.nencioni@unige.it

Reviewers' Comments:

Reviewer #1:

Remarks to the Author:

The authors have addressed my comments and improved the manuscript.

Reviewer #2:

Remarks to the Author:

My earlier concerns have been addressed appropriately in the revised paper I believe.

Reviewer #3:

Remarks to the Author:

The authors thoughtfully addressed the review comments with the exception that they reference a Supplemental Figure 4f showing rescue of Capan-1 xenografts by LDL administration. I was not able to find this figure.

Genoa, September 18th 2023

Reviewer #3 (Remarks to the Author): The authors thoughtfully addressed the review comments with the exception that they reference a Supplemental Figure 4f showing rescue of Capan-1 xenografts by LDL administration. I was not able to find this figure.

Indeed, the figure in which this experiment (“*rescue of Capan-1 xenografts by LDL administration*”) is shown is main figure 3e, which we had not mentioned in the text. So, we are thankful to the referee for having pointed this out. We have addressed this and in the resubmitted version of the article, the sentence from the main text, in which the experiment is described, is the following “... Similar to LDL cholesterol add-back, administering recombinant insulin, IGF1 and leptin to Capan-1 xenograft-bearing mice abolished fasting-induced enhancement of terbinafine activity (Fig. 3e, f) ...” (from the section “Regulation of tumour cholesterol metabolism by fasting”).

Thanks again for your consideration.

With my warmest regards,

Alessio Nencioni, MD

Email: alessio.nencioni@unige.it